# Efficient Prediction of pass@$k$ Scaling in Large Language Models

## Abstract

Assessing the capabilities and risks of frontier AI systems is a critical area of research, and recent work has shown that repeated sampling from models can dramatically increase both. For instance, repeated sampling has been shown to increase their capabilities, such as solving difficult math and coding problems, but it has also been shown to increase their potential for harm, such as being jailbroken. Such results raise a crucial question for both capability and safety forecasting: how can one accurately predict a model's behavior when scaled to a massive number of attempts, given a vastly smaller sampling budget? This question is directly relevant to model providers, who serve hundreds of millions of users daily, and to governmental regulators, who seek to prevent harms. To answer this questions, we make three contributions. First, we find that standard methods for fitting these laws suffer from statistical shortcomings that hinder predictive accuracy, especially in data-limited scenarios. Second, we remedy these shortcomings by introducing a robust estimation framework, which uses a beta-binomial distribution to generate more accurate predictions from limited data. Third, we propose a dynamic sampling strategy that allocates a greater budget to harder problems. Combined, these innovations enable more reliable prediction of rare risks and capabilities at a fraction of the computational cost.

## 1 Introduction

Prompt-based attacks against frontier (multimodal) AI systems often fail when attempted only once (Anil et al., 2024; Panfilov et al., 2025; Howe et al., 2025; Kazdan et al., 2025). Likewise, many hard math (Glazer et al., 2024) and software engineering (Jimenez et al., 2024) tasks are too difficult for models to solve reliably on the first attempt. Through repeated attempts, however, the success rate of these models can climb rapidly to near-100% (Brown et al., 2024; Hughes et al., 2024; Kwok et al., 2025). Consequently, predicting changes in capabilities and/or risks when a user is allowed many attempts to accomplish a task has become an important problem for companies, researchers, and governmental regulators alike. The relevance of this problem is only underscored by the massive scale at which these frontier AI systems are deployed, with some experiencing billions of daily interactions. However, making such predictions is challenging because sampling from language models at such scale can be prohibitively expensive. How can one predict the behavior of frontier AI systems in this repeated attempts regime using only a limited number of samples?

In this work, we approach this problem through estimation of the widely used pass@$k$ metric (Kulal et al., 2019; Chen et al., 2021), which measures the expected pass rate given $k$ attempts at solving each problem, where a problem is solved if any attempt is successful. Unfortunately, direct estimation at high $k$ is often difficult. While prior work has shown that pass@$k$ can follow predictable power laws across a range of domains including jailbreaking, mathematical problem-solving, and code generation (Hughes et al., 2024; Brown et al., 2024; Du et al., 2024), we find that standard methods for fitting these laws (Chen et al., 2021; Brown et al., 2024; Hughes et al., 2024) suffer from statistical shortcomings that hinder predictive accuracy, especially in data-limited scenarios.

We argue that the shortcomings of prior prediction methods stem from statistical approximations that do not hold in sample-limited regimes. By carefully modeling the data-generating process and developing faithful estimators, we demonstrate that predictions can be substantially improved.

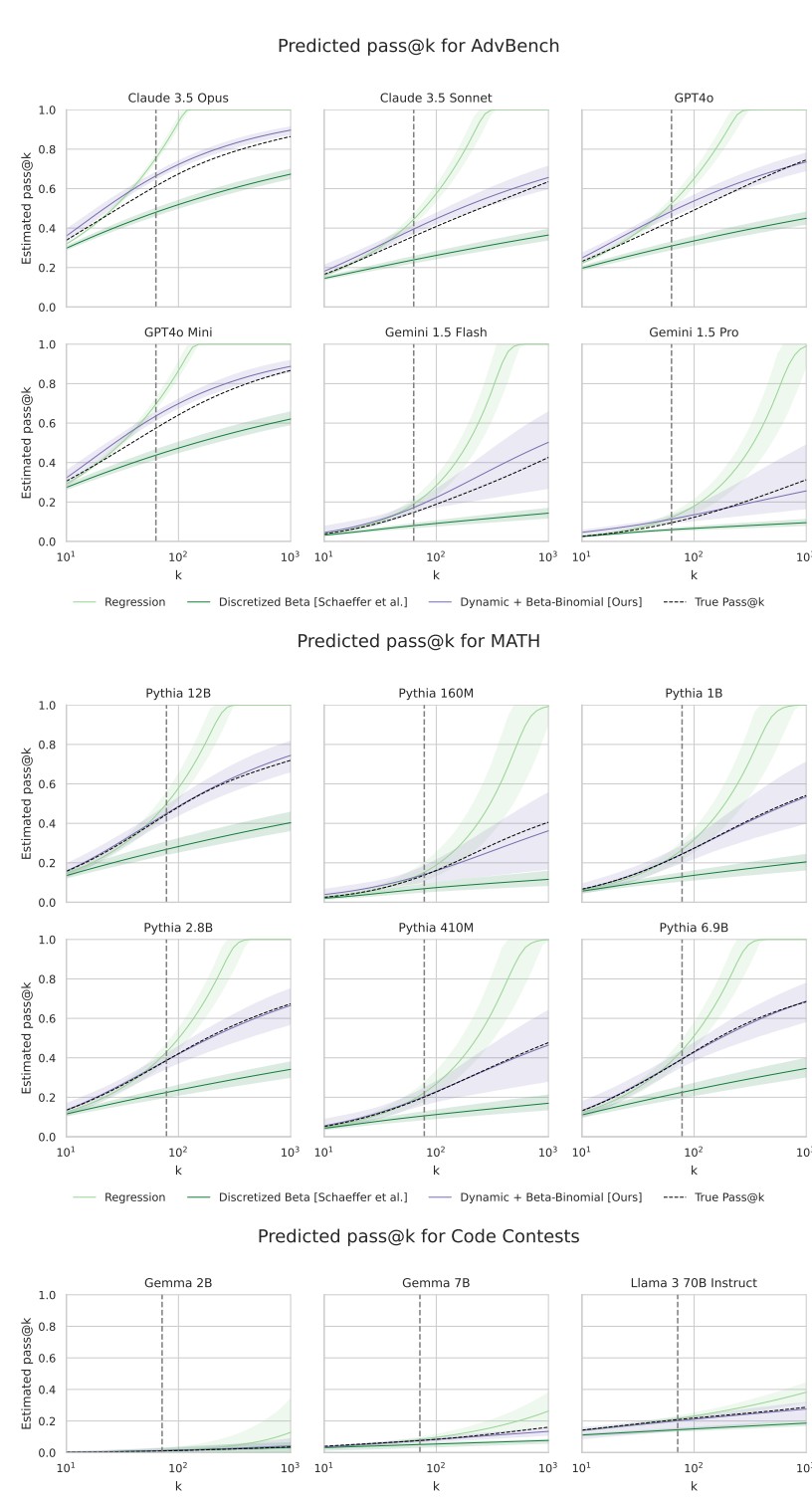

Figure 1: **Comparing Forecasting Methods for** $\text{pass}_{\mathcal{D}}@k$ **Across Different Datasets.** The ground truth is computed based on 10 000 actual samples per problem. All predictive models are trained on data from a budget of 10 000 total samples. **The gray region** shows $k$ for which $\text{pass}@k$ can be directly estimated given the available budget, while **the white region** shows $k$ for which the $\text{pass}@k$ must be extrapolated given the budget. Our estimator tracks the ground truth far better than prior methods. Error bars represent a bootstrapped 95% confidence interval.

## 1.1 CONTRIBUTIONS

To address the challenge of efficient prediction, this paper makes the following contributions:

1. **Rigorous critique of prior prediction methods.** We discuss statistical flaws that have led to poor prediction accuracy in common approaches such as log-log linear regression and existing distributional fitting techniques.

2. **Robust estimation framework for prediction.** We remedy the shortfalls of previous methods by employing a more suitable distributional model—the beta-binomial—and deriving an improved predictor for $\text{pass@}k$ that more faithfully accounts for the data generating process in order to deliver more accurate predictions.

3. **Efficient dynamic sampling strategy.** We show empirically that by allocating our fixed compute budget adaptively to focus on more difficult problems, we achieve more accurate predictions than the standard approach of uniform sampling.

The insights from this work are important for both AI safety and capabilities research. For AI safety, reliable forecasts for the scaling of vulnerability rates is crucial for assessing the societal risk posed by models deployed to millions of users. For capabilities, such predictions are vital for efficiently applying methods like Reinforcement Learning from Verified Rewards (RLVR), where training on difficult problems requires correctly sizing batches to ensure a non-zero success rate. Thus, efficiently predicting the scaling of risks and capabilities is a critical step towards developing aligned and powerful AI systems.

## 2 PROBLEM STATEMENT: EFFICIENT PREDICTION OF RARE MODEL BEHAVIORS FROM REPEATED SAMPLING

We consider the performance of AI systems on some problem, defined as a set of prompts with verifiable binary outcomes: each attempt either produces the (un)desirable outcome for that prompt, or does not. For example, we may want our AI system to solve a Millennium Problem, or to not launch a cyberattack on a nation's infrastructure. Our goal is to predict the success rate of an AI system, given many repeated attempts at the problem. To quantitatively measure the system's behavior, we use the widely-adopted "pass-at-k" metric (Kulal et al., 2019): For a single prompt, indexed by $i$, from a distribution of prompts $\mathcal{D}$, let $\text{pass}_i\text{@}1$ be the model's true probability of success in one attempt. The probability of achieving at least one success in $k$ attempts is then $\text{pass}_i\text{@}k$:

$$\text{pass}_i\text{@}k = 1 - (1 - \text{pass}_i\text{@}1)^k. \tag{1}$$

For the entire dataset $\mathcal{D}$ of $m$ problems, the overall pass rate $\text{pass}_{\mathcal{D}}\text{@}k$ is the expected fraction of problems solved within $k$ attempts:

$$\text{pass}_{\mathcal{D}}\text{@}k = \mathbb{E}_{i \sim \mathcal{D}}[\text{pass}_i\text{@}k]. \tag{2}$$

Our goal is to predict performance given many attempts using data from an economically feasible, small-scale experiment. This leads to our formal research question:

> *Given a total compute budget of $B$ samples to be distributed across a dataset $\mathcal{D}$ containing $m$ problems, how should one best allocate this budget and build a model to predict $\text{pass}_{\mathcal{D}}\text{@}k$ for $k \gg B/m$?*

In this work, we use a small budget (e.g., $B/m \in [10^0, 10^2]$) to predict performance for $\text{pass@}k$ at large scale (e.g., $k \in [10^1, 10^4]$). We evaluate predictions by comparing them against a ground truth estimate of $\text{pass@}k$ computed using a withheld dataset of $10\,000$ samples per problem. To evaluate performance, we compute mean squared error (MSE) relative to the ground truth $\text{pass@}k$ value.

The product of our contributions is an estimator that provides consistently more accurate predictions than existing methods (see Figure 1).

## 3 CRITIQUING PAST METHODS OF PREDICTING $\text{pass}@k$

We now examine past methods of predicting $\text{pass}@k$ scaling and identify their shortcomings.

### 3.1 COMBINATORIAL ESTIMATION

Directly measuring $\text{pass}_{\mathcal{D}}@k$ for a large $k$ is often computationally expensive. While unbiased estimators exist, such as that of Chen et al. (2021), they are only defined when the number of samples taken for each problem is greater than or equal to the number of attempts $k$. Given $b_i$ samples on problem $i$ with $s_i$ successes, this estimator is:

$$\widehat{\text{pass}_i@k} = 1 - \frac{\binom{b_i - s_i}{k}}{\binom{b_i}{k}}. \tag{3}$$

In this paper, we focus on the regime where $B/m < k < B$. As the size and quantity of benchmarks continues to grow, we may often find ourselves in such constrained contexts. Here, given that $k > B/m$, we cannot allocate the required minimum of $k$ samples for each of $m$ problems. This means the standard unbiased estimator (Equation 3) cannot be directly applied, so we must instead rely on extrapolation and predictive modeling.

### 3.2 LINEAR REGRESSION

The first and most common extrapolation of $\text{pass}@k$ uses linear regression (Brown et al., 2024; Hughes et al., 2024). Specifically, given $b$ samples per problem, one first estimates $\text{pass}_{\mathcal{D}}@k$) for $k$ between 1 and $b$ and then fits a least squares regression of the form:

$$-\log(\text{pass}_{\mathcal{D}}@k) \sim a\log(k) + c. \tag{4}$$

Fixing $C = e^{-c}$ corresponds to the power law:

$$\text{pass}_{\mathcal{D}}@k \sim C \cdot k^{-a}. \tag{5}$$

Explicitly, the regression loss takes the form:

$$\frac{1}{|\mathcal{D}|} \sum_{i \in \mathcal{D}} \left( -\log\left(\widehat{\text{pass}_{\mathcal{D}}@k}\right) - a\log(k) - c \right)^2. \tag{6}$$

There are several problems with this approach, leading to poor estimates of $\text{pass}@k$ for higher $k$ values as shown in Figure 1:

1. Estimates of $\text{pass}_{\mathcal{D}}@k$ are not independent for different $k$ when they are computed using the same dataset of samples.

2. Estimates of $\text{pass}_{\mathcal{D}}@k$ are not homoskedastic, i.e. they have different variances for each value of $k$.

3. $\text{pass}@k$ may not actually follow a power law for some datasets.

4. Power laws typically apply only for large values of $k$. Therefore, if the computation budget for sampling is not large, then non-leading terms can dominate, resulting in poor fits of the data.

To provide a concrete example of the fourth point, suppose that

$$1 - \text{pass}@k = \frac{A}{k^{\alpha}} + \frac{B}{k^{\beta}} \tag{7}$$

where $A \gg B$ but $\alpha > \beta$. For small values of $k$, the first term of Equation 7 dominates. However, for large values of $k$, the second term, which supplies the true asymptotic power law, dominates. If we lack a sufficient budget to observe samples for large $k$, then least squares will incorrectly fit to the first term. We quantify statements 1 and 2 more precisely with proofs in Appendix B.

Our work directly remedies these issues by moving away from regression on aggregate statistics, instead modeling the underlying distribution of problem difficulties.

### 3.3 DISCRETIZED-BETA DISTRIBUTIONAL FITTING

Schaeffer et al. (2025) use a variant of empirical Bayes to estimate pass@$k$ for high $k$. To describe their method, we first introduce some notation. As before, let $\mathcal{D}$ denote a data set of questions. Define $\mathcal{U}$ to be the distribution of per-problem success probabilities $\text{pass}_i@1$ for $i \in \mathcal{D}$:

$$\text{pass}_i@1 \sim \mathcal{U}, \quad i \in \mathcal{D}. \tag{8}$$

For the $i$-th question in our dataset, we observe $b$ samples, of which we say that $s_i$ are successful. Schaeffer et al. (2025) fit scaled beta distributions to $\widehat{\text{pass}_i@1} = \frac{s_i}{b}$ and leverage this distribution to estimate pass@$k$ in the following steps.

**Step 1: Fit the scale $\theta$.** Recall the probability density function of a scaled beta distribution:

$$\text{Beta}(p; \alpha, \beta, \theta) = \frac{1}{\text{Be}(\alpha, \beta)} \left(\frac{p}{\theta}\right)^{\alpha-1} \left(1 - \frac{p}{\theta}\right)^{\beta-1} \frac{1}{\theta}, \tag{9}$$

Schaeffer et al. (2025) provide the following estimate for the scale parameter $\theta$:

$$\hat{\theta} = \frac{b+1}{b} \max_{i \in \mathcal{D}} \left(\widehat{\text{pass}_i@1}\right). \tag{10}$$

They use this estimator because it resembles the uniformly minimum variance unbiased estimator (UMVUE) for the parameter $B$ of a uniform distribution $\text{Uniform}(0, B)$ Lehmann (1983). Unfortunately, the scaled beta distribution is not an exponential family distribution. In particular, the UMVUE for $\theta$ in a scaled beta distribution is unknown. As such, this is not a principled estimator for $\theta$. We provide details for how to estimate $\theta$ using a stabilized MLE in Appendix C, but we find empirically that using the scale parameter does not improve predictions.

**Step 2: Fit $\alpha$ and $\beta$ by discretizing.** Schaeffer et al. (2025) first divide the interval $(0, 1)$ into log-scale bins with endpoints $0 = e_0, e_1, ..., e_\ell = 1$, where the bin widths decrease ($e_i - e_{i-1} > e_{i+1} - e_i$). They then numerically compute the probability mass in each bin and fit $\alpha$ and $\beta$ by maximizing the multinomial likelihood over the number of problems whose estimated success rate falls into each bin. Specifically, if we assign the estimated probability:

$$A_i(\alpha, \beta, \theta) := \int_{e_i}^{e_{i+1}} \text{Beta}(p; \alpha, \beta, \theta) dp, \tag{11}$$

then Schaeffer et al. (2025) fit $\alpha$ and $\beta$ by optimizing

$$\arg\min_{\alpha, \beta} -\log \left(\prod_{i=1}^{\ell} A_i(\alpha, \beta, \theta)^{\sum_{j=1}^{m} \mathbf{1}\{\widehat{\text{pass@1}} \in [e_i, e_{i+1})\}}\right) \tag{12}$$

$$= \arg\min_{\alpha, \beta} -\sum_{i=1}^{\ell} \left(\sum_{j=1}^{m} \mathbf{1}\{\widehat{\text{pass@1}} \in [e_i, e_{i+1})\}\right) \log\left(A_i(\alpha, \beta, \theta)\right). \tag{13}$$

This more complex discretized beta estimator was used to support the common case when $s_i = 0$. Here, the estimate $\widehat{\text{pass}_i@1}$ is also 0, meaning the scaled beta density is not supported.

**Step 3: Predict** pass@$k$    Schaeffer et al. (2025) use the fit distribution to approximate the asymptotic slope of the pass@$k$ scaling curve and do not attempt to extrapolate pass@$k$ beyond the provided number of trials. To extend this approach to the high-$k$ regime, we take the expectation over the success probability $\text{pass}_i@1 \sim \text{Beta}(\hat{\alpha}, \hat{\beta}, \hat{\theta})$:

$$\widehat{\text{pass}_i@k} = \mathbb{E}_{\text{pass}_i@1 \sim \text{Beta}(\hat{\alpha}, \hat{\beta}, \hat{\alpha})} \left[1 - (1 - \text{pass}_i@1)^k\right]. \tag{14}$$

**Analysis of the Discretized-Beta Estimator**    Because the bins are wider for smaller values, this fitting method consistently produces **downward-biased** estimates of the distribution $\mathcal{U}$. We demonstrate this phenomenon in Figure 2 where the discretized beta distribution is fit on problem success probabilities drawn from a uniform distribution. The fit is visibly skewed, incorrectly up-weighting the left tail of the distribution.

# 4    BETTER ESTIMATION OF $\mathrm{pass}@k$

In this section, we develop a novel predictor of $\mathrm{pass}_{\mathcal{D}}@k$ that achieves far better predictive accuracy for large $k$. We take inspiration from Levi (2024), who uses similar methods to model $\mathrm{pass}@k$. As shown in Figure 5, our method provides equivalent or better estimates across all models, values of $k$, and sampling budgets tested. We no longer assume a fixed sampling budget per question, so we denote the budget for the $i$-th question by $b_i$. Our improvements involve two steps:

1. We develop an alternative distributional fitting method for the problem-difficulty distribution $\mathcal{U}$.

2. We propose a simple dynamic sampling strategy to allocate the sample budget more efficiently.

## 4.1    FITTING THE PROBLEM-DIFFICULTY DISTRIBUTION $\mathcal{U}$

We denote the underlying distribution of per-problem success probabilities as $\mathrm{pass}_i@1 \sim \mathcal{U}$, where $\mathcal{U}$ is unknown. The number of successes $s_i$ on the $i$-th problem out of $b_i$ attempts is then binomially distributed: $s_i \sim \mathrm{Binomial}(b_i, \mathrm{pass}_i@1)$.

Instead of the biased discretization approach, we model $\mathcal{U}$ as a beta distribution. This allows us to leverage the properties of conjugate priors and fit a beta-binomial distribution directly to the observed counts of successes and trials $(s_i, b_i)$. The likelihood for the beta-binomial is given by:

$$\Pr\left[s = s_i \mid b = b_i; \alpha, \beta\right] = \binom{b_i}{s_i} \frac{\mathrm{Be}(s_i + \alpha, b_i - s_i + \beta)}{\mathrm{Be}(\alpha, \beta)}, \tag{15}$$

where $\mathrm{Be}(\cdot, \cdot)$ is the beta function. As shown in Figure 2, the discretized estimator badly fits a uniform distribution because it incorrectly puts excessive weight on the left tail. We also observe here the superior fit achieved by maximizing the beta-binomial likelihood directly, which ultimately results in better predictions of $\mathrm{pass}@k$.

Next, we obtain a maximum likelihood estimate for $\mathcal{U}$:

$$\hat{\alpha}, \hat{\beta} = \arg \max_{\alpha, \beta > 0} \prod_{i=1}^{m} \Pr\left[s = s_i \mid b = b_i; \alpha, \beta\right]. \tag{16}$$

Finally, we retrieve an estimate for $\mathrm{pass}@k$:

$$\widehat{\mathrm{pass}_i@k} = \mathbb{E}_{\mathrm{pass}_i@1 \sim \mathrm{Beta}(\hat{\alpha}, \hat{\beta})} \left[1 - (1 - \mathrm{pass}_i@1)^k\right]. \tag{17}$$

We see in Figure 2 that our approximate Beta-Bernoulli distribution better fits problem success probabilities sampled from a uniform distribution.

## 4.2    MORE EFFICIENT SAMPLING STRATEGIES

It was demonstrated by Schaeffer et al. (2025) that in the high-$k$ regime, $\mathrm{pass}@k$ scaling is governed almost exclusively by the shape of the difficulty distribution near 0. Distinguishing between an easy problem ($\mathrm{pass}_i@1 = 0.25$) and a very easy problem ($\mathrm{pass}_i@1 = 0.75$) provides little to no information. Therefore, we propose to concentrate our sampling budget on the hardest problems by always sampling responses to the hardest problem with the least number of attempts so far. We provide our dynamic problem selection criteria in Algorithm 1. Although we present the algorithm with fixed-length arrays for fixed datasets, there is a clear extension to infinite question pools when questions arrive in a stream.

---

**Algorithm 1** `SelectHardestProblem`

---

**Require:** Dataset $\mathcal{D}$ with $m$ problems and per-problem counts of successful and total attempts: `successes` and `attempts`, respectively.

$s^* \leftarrow \min_i \texttt{successes}_i$

$H \leftarrow \arg\min_{\{i \, : \, \texttt{successes}_i = s^*\}} \texttt{attempts}_i$

$i^* \sim \text{Uniform}(H)$

**return** $i^*$

---

This adaptive approach is not immediately applicable to the regression-based estimator, which requires a uniform number of samples across problems to compute intermediate $\text{pass}_{\mathcal{D}}@k$ values. It is likewise inconsistent with the discretized estimator from Schaeffer et al. (2025) since direct estimates $\hat{p}_i = \frac{s_i}{b_i}$ have different precision with this dynamic sampling method. However, our distributional fitting method remains valid, as the beta-binomial likelihood (Equation 15) can handle variable numbers of trials ($b_i$) for each problem. We outline our complete approach in Algorithm 2.

---

**Algorithm 2** Dynamic Sampling + Beta-Binomial Fit for Efficient $\text{pass}_{\mathcal{D}}@k$ Estimation

---

**Require:** Dataset $\mathcal{D}$ with $m$ problems, total sample budget $B$, and number of repeated attempts $k$.

Initialize $\texttt{successes}_i \leftarrow 0$ and $\texttt{attempts}_i \leftarrow 0$ for all $i \in \{1, \dots, m\}$

**for** $t \in \{1, \dots, B\}$ **do**

  $i_t \leftarrow \texttt{SelectHardestProblem}(s, b)$

  $\texttt{attempts}_{i_t} \leftarrow \texttt{attempts}_{i_t} + 1$

  $\texttt{successes}_{i_t} \leftarrow \texttt{successes}_{i_t} + \mathbf{1}\{\texttt{AttemptProblem}(i_t)\}$

**end for**

$\hat{\alpha}, \hat{\beta} \leftarrow \arg\max_{\alpha, \beta > 0} \prod_{i=1}^{m} \Pr\left[s = s_i \mid b = b_i; \alpha, \beta\right]$        Equation 16

$\widehat{\text{pass}_i@k} \leftarrow \mathbb{E}_{\text{pass}_i@1 \sim \text{Beta}(\hat{\alpha}, \hat{\beta})}\left[1 - (1 - \text{pass}_i@1)^k\right]$        Equation 17

**return** $\widehat{\text{pass}_i@k}$

---

**On improved sample allocation.** The decision to select problems dynamically based on estimated problem difficulty is motivated by intuition from the theorems in Schaeffer et al. (2025). It is generally difficult to analyze the effect of such adaptive schemes in a Bayesian context. Therefore, to provide theoretical motivation for our approach, we introduce a natural frequentist estimator, defined below. Given oracle access to $\text{pass}_i@1$ and control over the number of samples taken for each problem $b_i$, we prove that the variance of this estimator can be minimized by prioritizing "harder" problems with low $\text{pass}_i@1$.

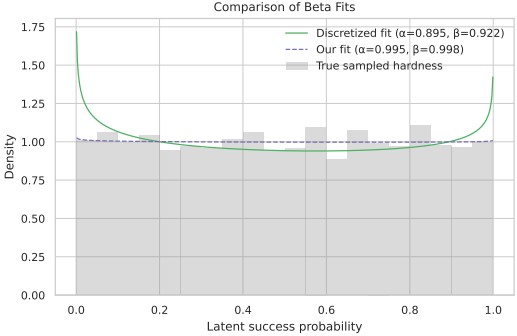

Figure 2: **Comparing Hardness Distribution Fit for Discretized Beta vs. Beta-Bernoulli**. $m = 10\,000$ problem success probabilities are sampled: $\text{pass}_i@1 \sim \text{Uniform}([0, 1])$. $b = 100$ success/failure samples are drawn for each problem, $s_i \sim \text{Bin}(b, \text{pass}_i@1)$.

**Theorem 1.** *Consider the following frequentist estimator of* $\mathrm{pass}@k$

$$\widehat{\mathrm{pass}_i@k}_{freq} := 1 - \frac{1}{n} \sum_{i=1}^{n} (1 - s_i/b_i)^k.$$

*In the asymptotic regime as* $n \to +\infty$, *the sampling budget* $b^*$ *that minimizes the variance* $\mathrm{Var}(\widehat{\mathrm{pass}_i@k}_{freq})$ *is:*

$$b_i^* \propto \sqrt{(\mathrm{pass}_i@1)(1 - \mathrm{pass}_i@1)^{2k-1}}.$$

A proof of Theorem 1 is provided in Appendix E. The result further motivates our use of dynamic sampling. We conjecture that such adaptive strategies can also reduce variance in the context of our multi-stage Bayesian approach, but we leave such detailed analysis for future work.

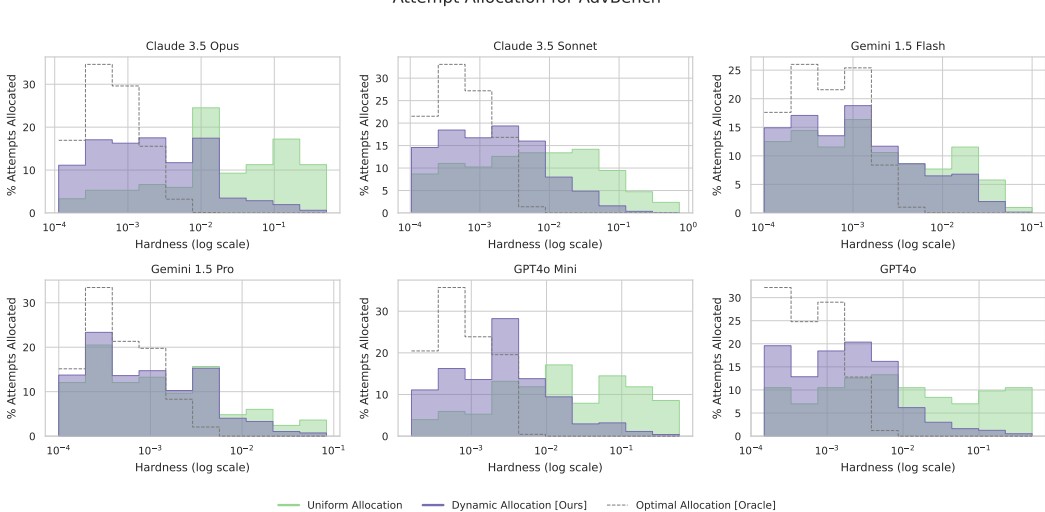

Figure 3: **Budget Allocation by Hardness Relative to the Optimal Allocation from Theorem 1** Contrasted distributions of problem success probabilities for the problems selected by dynamic and uniform sampling strategies on AdvBench. Note that these probabilities are not immediately available to our estimator but rather approximated given a limited amount of samples for each problem. The dotted line represents the distribution of problem success probabilities under the optimal sampling allocation provided in Theorem 1, assuming oracle access to the problem success probabilities. We see that the dynamic strategy is more closely aligned with this optimal rate.

Beyond this, we show in Figure 3 that the distribution of the difficulties of problems selected by our dynamic strategy aligns much more closely with the derived optimal allocation from Theorem 1 than that of the uniform strategy.

However, in the sample-count regimes and distributions in our datasets, it is difficult to empirically isolate the benefits of the sampling method alone. Therefore, we provide some additional empirical support for dynamic sampling on synthetic data in Appendix E. We find that when there are many easy problems and a small number of hard outliers, or a uniform distribution of difficulties, the dynamic sampling method outperforms uniform sampling by large margins. On all distributions tested, dynamic sampling performs better than or comparably to uniform sampling.

## 5 RESULTS

In this section, we evaluate the predictive accuracy of our method against prior work. We estimate $\mathrm{pass}_{\mathcal{D}}@k$ for $k$ in the range $[10^1, 10^3]$ on three real-world datasets and three to six different models for each dataset

## 5.1 Experimental Setup

We source our data from Brown et al. (2024) and Hughes et al. (2024), which contain $10\,000$ sampled successful or failed attempts for each of $100 \sim 200$ problems selected from Code Contests (Li et al., 2022), MATH (Hendrycks et al., 2021), and AdvBench (Zou et al., 2023).

For model fitting, we use a budget of $10^1 < B < 10^4$ samples.

- For methods requiring uniform sampling (Log-Log Regression, Discretized Beta), we shuffle the samples within each problem and use the first $B/m$ for each problem.
- For our primary method (Dynamic Sampling + Beta-Binomial Fit) we again use the shuffled data but instead run our estimator, defined in Algorithm 2.

We predict $k$ between $100$ and $10\,000$, with $k$ chosen spaced on a log scale and compute squared error. Ground truth estimates are computed for $\text{pass}@k$ using all $10\,000$ available samples.

## 5.2 Discussion

The predictions for AdvBench, MATH, and Code Contests with different sampling budgets are shown in Figure 1. We observe that **existing estimators diverge significantly from the true** $\text{pass}@k$ **value beyond this threshold**. Figure 5 provides a heat map of errors for different sampling budgets and values of $k$. Note that, as expected, the error decreases as we increase the sampling budget. Existing estimators especially struggle with high values of $k$. We also provide the MSE for each estimator across different sampling budgets in Appendix F.

Across models and datasets, our proposed method provides predictions that are closest to the ground truth. The predictions from log-log regression are particularly poor, often diverging to predict impossible pass rates greater than 1 (we clip these at 1 for visualization and error computation). The prior distributional fitting method from Schaeffer et al. (2025) performs better than unclipped regression but consistently underestimates $\text{pass}@k$ for large $k$.

The gains from our method come from several sources. Our fitting uses maximum likelihood estimation on an exponential family model, which is known to have properties like asymptotic normality, unbiasedness, and $O(1/\sqrt{n})$-convergence. We avoid the pitfall of fitting models on top of correlated estimates, as in the regression method. Finally, we align our sampling budget more closely with the distribution that theoretically minimizes variance.

## 6 Conclusion and Future Work

Predicting the capabilities and vulnerabilities of AI models at scale is a critical challenge. We contribute to more efficient and accurate prediction by making two core improvements: (1) selecting a more appropriate model for the underlying problem difficulties, and (2) utilizing dynamic sampling to concentrate compute on the most difficult problems. We demonstrate the significant impact of these innovations in Figure 5 on mathematical problem-solving.

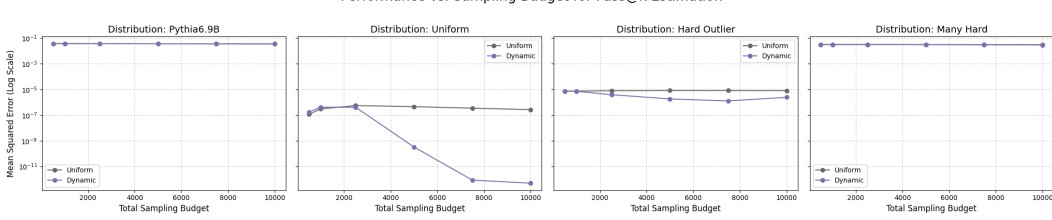

Figure 4: **Evaluating Performance Scaling for Uniform vs. Dynamic Allocation Strategies** Dynamic sampling is most useful when there are a handful of very difficult problems, but many easy problems. These distributions allow it to concentrate a large proportion of the budget on difficult problems. The "Hard Outlier" distribution has a single very difficult problem with success probability $1e-4$, and all other problems with difficulties in the range of $0.1$-$0.3$.

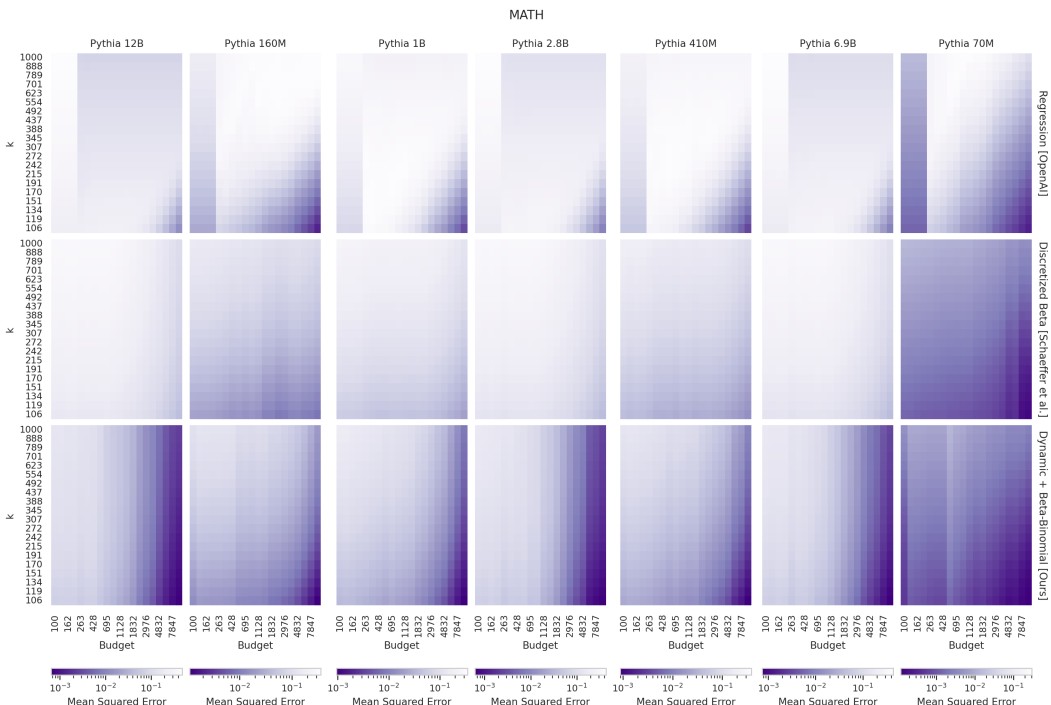

Figure 5: Heatmap depicting how predictions of $\text{pass}@k$ change with the sampling budget and $k$ on MATH. Our method minimizes MSE for virtually all values of $k$ and sampling budgets, as evidenced by the darker colors in its heatmap. Figures for MATH and Code Contests are in Appendix F.

We achieved large improvements in predictive accuracy by remedying statistical errors in prior methods and improving sampling techniques, without requiring extra sampling compute. These gains suggest that a closer statistical inspection of other scaling-law fitting methodologies could lead to considerable computational savings and, ultimately, better and safer models.

# 7 RELATED WORK

Repeated sampling from LMs improves performance on verifiable tasks, and has become the backbone of reinforcement learning from verified rewards Shao et al. (2024). The HumanEval benchmark first defined pass@$k$ and derived an unbiased estimator for this quantity (Chen et al., 2021). Follow-up work on code generation studied sampling as a resource that must be allocated across problems; for instance, Han et al. (2023) adaptively prioritizes sampling solvable instances. They estimate pass@$k$ with Monte-Carlo, whereas we predict how pass@$k$ varies with $k$ on a fixed budget.

For rewards that are not easily verifiable, repeated sampling must be combined with aggregation techniques such as majority voting (Wang et al., 2023). Self-consistency shows that majority voting over multiple chains-of-thought can boost reasoning performance, and adaptive-consistency (Aggarwal et al., 2023) reduces compute by stopping sampling once a clear majority emerges. Chen et al. (2024) study compound AI systems that repeatedly query an LM and aggregate responses via vote or filter-vote, and they describe how performance varies with the number of LM calls.

Statistically, estimating pass@$k$ involves binomial success probabilities that can be extremely small on difficult benchmark items, making variance control and uncertainty quantification important. Classical work on binomial proportion intervals compares exact and approximate methods, including the Clopper-Pearson interval and score-based or adjusted Wald intervals (Brown et al., 2001). Recent analysis in the rare-event regime emphasizes that both coverage and relative margin of error are important when designing estimators and determining sample sizes McGrath & Burke (2024).

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

## A  LIMITATIONS

While we predict pass@$k$ on a scale of pass@10 000 using dozens to hundreds of samples, frontier labs face the challenge of predicting pass@$k$ for $k$ several orders of magnitude higher. These frontier labs also have the capability to generate far more samples to aid their predictions than we can. Although we hope that the lessons of how to cross orders of magnitude when predicting pass@$k$ transfer, power law fitting might be more viable when the number of samples available is higher. In the same vein, because we are limited in the number of samples that we can draw, we are unable to estimate the precision of pass@1 beyond $1e - 4$, which creates hard limits for us when testing the accuracy of our models.

## B  PITFALLS OF LINEAR REGRESSION

In this section, we precisely quantify the statements made in Section 3.2.

**The estimates $\widehat{\text{pass}@k}$ are not independent for different** $k$**:** Recall that one of the assumptions of the linear regression model is that the observations are independent. The following lemma characterizes this non-independence on a per-problem basis:

**Lemma 1.** *Recall that $s_i$ is the number of successes observed out of $b$ attempts on the $i$th problem of $\mathcal{D}$. If $k \geq l$, and $0 < s_i < b$ then there exists an invertible function $f$ such that*

$$\widehat{pass_i@k} = f\left(\widehat{pass_i@l}\right). \tag{18}$$

*This invertible function takes the form:*

$$f\left(\widehat{pass_i@l}\right) = \widehat{pass_i@l} + s_i \sum_{m=l}^{k-1} \frac{\binom{b-s_i}{m}}{(b-m)\binom{b}{m}}. \tag{19}$$

*Proof.*

$$\text{Let} \quad g(m) = \frac{\binom{b-s_i}{m}}{\binom{b}{m}}, \quad \text{then} \quad \widehat{pass_i@m} = 1 - g(m).$$

Now, $\quad \dfrac{g(m+1)}{g(m)} = \dfrac{\binom{b-s_i}{m+1}}{\binom{b}{m+1}} \cdot \dfrac{\binom{b}{m}}{\binom{b-s_i}{m}}$

$$= \frac{\binom{b-s_i}{m+1}}{\binom{b-s_i}{m}} \cdot \frac{\binom{b}{m}}{\binom{b}{m+1}}$$

$$= \frac{b - s_i - m}{m+1} \cdot \frac{m+1}{b-m}$$

$$= \frac{b - s_i - m}{b - m}.$$

$$\Rightarrow 1 - g(m+1) = 1 - \frac{b - s_i - m}{b - m} g(m)$$

$$\Rightarrow 1 - g(m+1) = (1 - g(m)) + g(m)\left(1 - \frac{b - s_i - m}{b - m}\right)$$

$$= (1 - g(m)) + g(m) \cdot \frac{s_i}{b - m}.$$

$$\Rightarrow \widehat{\text{pass}_i @(m+1)} = \widehat{\text{pass}_i @ m} + g(m) \cdot \frac{s_i}{b - m}$$

$$\Rightarrow \widehat{\text{pass}_i @ k} = \widehat{\text{pass}_i @ l} + s_i \sum_{m=l}^{k-1} \frac{1}{b - m} g(m) \quad \text{as desired.}$$

$\square$

This lemma implies that given $\text{pass}_i @ k$ for any $k$, $\text{pass}_i @ j$ for $j \neq k$ is uniquely determined.

**The estimates of $\widehat{\text{pass}@k}$ have different variances for different values of $k$:** A second assumption of the linear regression model is that the noise in the model is homoscedastic, i.e. the noise is the same for all $k$. This is again not the case for the estimators $\widehat{\text{pass}@k}$. The following lemma gives one instance in which these estimators are not homoscedastic:

**Lemma 2.** *Suppose that we have $n$ samples from a language model on problem $i$, and the language model has true probability $p$ of getting problem $i$ correct. Then*

$$Var\left(\widehat{\text{pass}_i @ n}\right) = (1 - p)^n - (1 - p)^{2n}, \tag{20}$$

*and*

$$Var\left(\widehat{\text{pass}_i @ 1}\right) = p(1 - p)/n. \tag{21}$$

*Proof.* Let $c \sim \text{Binomial}(n, p)$ be the number of correct completions obtained from $n$ i.i.d. samples of a fixed problem $i$. For each $k \in \{0, 1, \ldots, n\}$ define the empirical *pass@k* estimator

$$\widehat{\text{pass}_i @ k} = f_k(c), \text{ where } f_k(c) = 1 - \frac{\binom{n-c}{k}}{\binom{n}{k}}$$

Our goal is to show that the variances of $\widehat{\text{pass}_i @ k}$ are not constant in $k$. We begin with the variance in its raw definition:

$$\text{Var}\big[f_k(c)\big] = \underbrace{\mathbb{E}\big[f_k(c)^2\big]}_{(a)} - \underbrace{\Big(\mathbb{E}\big[f_k(c)\big]\Big)^2}_{(b)}. \tag{$\star$}$$

Both expectations can be written as finite sums over the binomial probability-mass function:

$$(a) = \sum_{c=0}^{n}\left(1 - \frac{\binom{n-c}{k}}{\binom{n}{k}}\right)^2 \binom{n}{c} p^c (1-p)^{n-c}, \quad (b) = \left(\sum_{c=0}^{n}\left(1 - \frac{\binom{n-c}{k}}{\binom{n}{k}}\right)\binom{n}{c} p^c (1-p)^{n-c}\right)^2.$$

We now specialize to two extreme choices of $k$.

CASE $k = n$

Because $\binom{n-c}{n} = 1$ if $c = 0$ and 0 otherwise,

$$f_n(c) = 1 - \binom{n-c}{n} = \mathbf{1}_{\{c \geq 1\}} \in \{0, 1\}, \quad \text{hence } f_n(c)^2 = f_n(c).$$

Next we compute the first and second moments.

$$\mathbb{E}[f_n(c)] = \mathbb{E}[f_n(c)^2] = \sum_{c=0}^{n} \mathbf{1}_{\{c \geq 1\}} \binom{n}{c} p^c (1-p)^{n-c}$$

$$= \sum_{c=1}^{n} \binom{n}{c} p^c (1-p)^{n-c}$$

$$= 1 - \binom{n}{0} p^0 (1-p)^n, \quad \text{Since the binomial PMF is normalized}$$

$$= 1 - (1-p)^n$$

Plugging the two moments into $(\star)$,

$$\text{Var}\big[f_n(c)\big] = \big[1 - (1-p)^n\big] - \big[1 - (1-p)^n\big]^2 = (1-p)^n - (1-p)^{2n}.$$

CASE $k = 1$

$$f_1(c) = 1 - \frac{n-c}{n} = \frac{c}{n}.$$

Because $\mathbb{E}[c] = np$ and $\text{Var}[c] = np(1-p)$,

$$\mathbb{E}[f_1(c)] = \frac{1}{n} \mathbb{E}[c] = p, \quad \text{and}$$

$$\mathbb{E}\big[f_1(c)^2\big] = \frac{1}{n^2} \mathbb{E}[c^2]$$

$$= \frac{1}{n^2} \big(\text{Var}[c] + \mathbb{E}[c]^2\big)$$

$$= \frac{1}{n^2} \big(np(1-p) + n^2 p^2\big)$$

$$= \frac{p(1-p)}{n} + p^2.$$

finally,

$$\text{Var}\big[f_1(c)\big] = \Big(\frac{p(1-p)}{n} + p^2\Big) - p^2 = \frac{p(1-p)}{n}. \qquad \square$$

## C MORE FLEXIBLE FITTING METHODS

Schaeffer et al. (2025) claimed that a standard beta distribution was not flexible enough to fit the distribution of $\text{pass}_i@1$, leading them to model the distribution of $\text{pass}_i@k$ as a scaled beta-binomial rather than a beta-binomial distribution. The authors developed the discretized fitting method described in Section 3.3 because they could not find a tractable likelihood for the three-parameter beta-binomial distribution.

In this section, we derive a tractable likelihood for the scaled beta-binomial distribution, allowing us to avoid estimating $\hat{\theta}$ from equation 9 using the unprincipled estimator from equation 10. A tractable likelihood also allows us to fit the scaled beta-binomial distribution directly to $n, k_i$ rather than first estimating $\text{pass}_i@k$ and fitting the scaled beta distribution to these estimates.

We first rewrite the expression for the likelihood of the scaled beta-binomial distribution to remove the integral in the following lemma:

**Lemma 3.** *The likelihood for the scaled beta-binomial distribution is given by*

$$\frac{1}{\text{Be}(\alpha,\beta)}\binom{n}{k}\int_0^\theta p^k(1-p)^{n-k}\left(\frac{p}{\theta}\right)^{\alpha-1}\left(1-\frac{p}{\theta}\right)^{\beta-1}\frac{1}{\theta}dp \tag{22}$$

$$=\frac{1}{\text{Be}(\alpha,\beta)}\binom{n}{k}\sum_{i=0}^{n-k}\binom{n-k}{i}(-1)^i\theta^{k+i}\text{Be}(k+i+\alpha,\beta). \tag{23}$$

The proof can be found in Appendix D.

Although the resulting likelihood no longer contains an integral, it involves an alternating sum of potentially large terms. Define

$$W_i=\binom{n-k}{i}\theta^{k+i}\text{Be}(k+i+\alpha,\beta). \tag{24}$$

In terms of $W_i$, our optimization objective is

$$-\log\left(\sum_{i=0}^{n-k}(-1)^iW_i\right). \tag{25}$$

To compute this as stably as possible, we use an alternating log-sum-exp function. Letting $W_m=\max\{W_0,...,W_{n-k}\}$, our log likelihood becomes:

$$-\log\left(\sum_{i=0}^{n-k}(-1)^i\exp(\log(W_i)-\log(W_m))\right)-\log(W_m). \tag{26}$$

## D    SCALED BETA-BINOMIAL LIKELIHOOD

$$\frac{1}{\text{Be}(\alpha,\beta)}\binom{n}{k}\int_0^\theta p^k(1-p)^{n-k}\left(\frac{p}{\theta}\right)^{\alpha-1}\left(1-\frac{p}{\theta}\right)^{\beta-1}\frac{1}{\theta}dp \tag{27}$$

$$=\frac{1}{\text{Be}(\alpha,\beta)}\binom{n}{k}\theta^k\int_0^\theta\left(\frac{p}{\theta}\right)^k(1-p)^{n-k}\left(\frac{p}{\theta}\right)^{\alpha-1}\left(1-\frac{p}{\theta}\right)^{\beta-1}\frac{1}{\theta}dp \tag{28}$$

$$=\frac{1}{\text{Be}(\alpha,\beta)}\binom{n}{k}\theta^k\sum_{i=0}^{n-k}\binom{n-k}{i}\int_0^\theta(-1)^ip^i\left(\frac{p}{\theta}\right)^{k+\alpha-1}\left(1-\frac{p}{\theta}\right)^{\beta-1}\frac{1}{\theta}dp \tag{29}$$

$$=\frac{1}{\text{Be}(\alpha,\beta)}\binom{n}{k}\sum_{i=0}^{n-k}\binom{n-k}{i}\int_0^\theta\theta^{k+i}(-1)^i\left(\frac{p}{\theta}\right)^{k+i+\alpha-1}\left(1-\frac{p}{\theta}\right)^{\beta-1}\frac{1}{\theta}dp \tag{30}$$

$$=\frac{1}{\text{Be}(\alpha,\beta)}\binom{n}{k}\sum_{i=0}^{n-k}\binom{n-k}{i}(-1)^i\theta^{k+i}\text{Be}(k+i+\alpha,\beta) \tag{31}$$

$$=\frac{1}{\text{Be}(\alpha,\beta)}\binom{n}{k}\sum_{i=0}^{n-k}\binom{n-k}{i}(-1)^i\theta^{k+i}\text{Be}(k+i+\alpha,\beta) \tag{32}$$

Define

$$W_i=\binom{n-k}{i}\theta^{k+i}\text{Be}(k+i+\alpha,\beta). \tag{33}$$

Our optimization objective is

$$-\log\left(\sum_{i=0}^{n-k}(-1)^iW_i\right). \tag{34}$$

To compute this as stably as possible, we use an alternating log-sum-exp function. Letting $W_m = \max\{W_0, ..., W_{n-k}\}$, our log likelihood becomes:

$$-\log\left(\sum_{i=0}^{n-k}(-1)^i \exp(\log(W_i) - \log(W_m))\right) - \log(W_m). \tag{35}$$

$$\text{pass}_i@1 \sim \text{Beta}(\alpha, \beta, \theta)$$
$$k_i \sim \text{Binomial}(n, \text{pass}_i@1)$$

## E  OPTIMAL DISTRIBUTION OF SAMPLES

### E.1  PROOFS

**Lemma 4** (Variance in the Asymptotic Regime). *For a sequence of random random variables $\{x_n\}$ such that $x_n = y_n/n$ where $y_n \sim Bin(n, p)$, we have the following:*

$$\sqrt{n}((1 - x_n)^k - (1 - p)^k) \overset{d}{\to} \mathcal{N}(0, pk^2(1 - p)^{2k-1})$$

*Proof.* By the Central Limit Theorem,

$$\sqrt{n}((1 - x_n) - (1 - p)) \overset{d}{\to} \mathcal{N}(0, p(1 - p)) \tag{36}$$

Let $g : \mathbb{R} \to \mathbb{R}$ be defined as follows:

$$g(t) = t^k$$

Applying the delta method:

$$\sqrt{n}((1 - x_n)^k - (1 - p)^k) \overset{d}{\to} \mathcal{N}(0, g'(1 - p)^2 p(1 - p)) \tag{37}$$

$$\overset{d}{\to} \mathcal{N}(0, (k(1 - p)^{k-1})^2 p(1 - p)) \tag{38}$$

$$\overset{d}{\to} \mathcal{N}(0, pk^2(1 - p)^{2k-1}) \tag{39}$$

$\square$

**Lemma 5** (Variance-Minimizing Budget). *Consider a random variable $X = \sum_{i=1}^m X_i$ where each $X_i$ is an independent random variable with variance $\text{Var}(X_i) = v_i/b_i$.*

*Consider the positive scaled simplex $B = \{b : b_i > 0 \ \& \ \sum_j^m b_j = B\}$. We have the following:*

$$b^* = \arg\min_{b \in B} \text{Var}(X; b) \tag{40}$$

$$b_i^* = \frac{\sqrt{v_i}}{\sum_j^m \sqrt{v_j}} \tag{41}$$

*Proof.* Our objective is this:

$$\min_{b_i > 0} \sum_{i=1}^m v_i/b_i \ \text{s.t.} \ \sum_{i=1}^m b_j = B$$

This objective is convex as a sum of convex functions, meaning we can use the Lagrange method:

$$\mathcal{L}(b, \lambda) = \sum_{i=1}^m v_i/b_i + \lambda\left(\sum_{i=1}^m b_i - B\right) \tag{42}$$

Applying first order conditions we get the following:

$$\frac{\partial \mathcal{L}}{\partial b_i} = -v_i/b_i^2 + \lambda \tag{43}$$

$$0 = -v_i/b_i^2 + \lambda \tag{44}$$

$$b_i = \sqrt{v_i/\lambda} \tag{45}$$

$$b_i \propto \sqrt{v_i} \tag{46}$$

$\square$

Combining Lemma 4 and Lemma 5, we have Theorem 1.

### E.2 SYNTHETIC COMPARISON OF UNIFORM AND DYNAMIC SAMPLING

We demonstrate the gains possible with dynamic sampling via the following contrived scenario: half of the problems are "easy" ($\text{pass}_i@1 = 0.3$) and half of the problems are "impossible" ($\text{pass}_i@1 = 0$). In this instance, we expect $\text{pass}@k \to 1/2$ as $k \to \infty$. However, without a sufficient allocation of samples to the "impossible" problems, the uniform sampling strategy prevents our estimator from determining whether these problems are impossible or just hard (i.e., still likely to be solved in $k$ attempts). This results in an upwards-biased estimate and relatively slow improvement of MSE as the budget grows. We observe this play out in Figure 7.

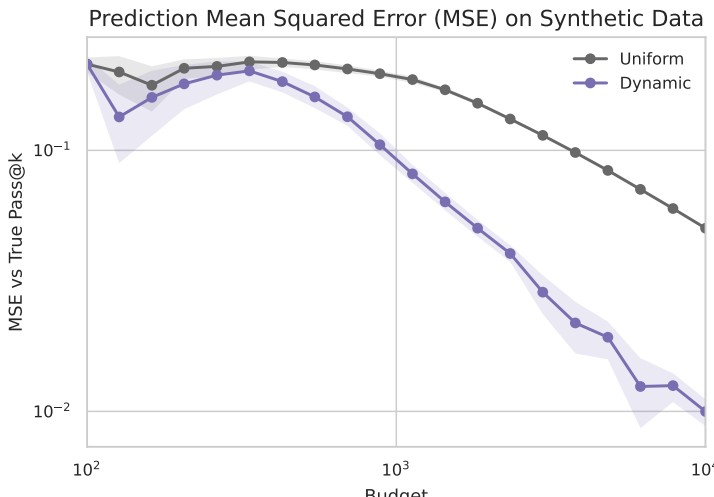

Figure 6: The MSE of our estimator with both dynamic and uniform sampling strategies given the described synthetic problem success probabilities, $n = 100$ problems and $k = 1\,000$. By focusing on the most difficult problems, the dynamic strategy allows our estimator to converge rapidly to the true $\text{pass}@k$ value.

We also provide some insight into the distributions for which dynamic sampling has advantages over uniform. We find that for uniform difficulty distributions or distributions that contain a handful of very hard outlier problems, dynamic sampling provides the most advantage. For distributions with many (or mostly) difficult problems, dynamic sampling holds little to no advantage over uniform sampling, since in these cases, uniform and dynamic sampling distribute the budget very similarly. If only a handful of problems are quickly solved, then dynamic sampling has very little extra samples to allocate to the more difficult problems.

| Method | Uniform Sampling | Dynamic Sampling |
|---|---|---|
| Beta-Binomial | 0.000017 [0.000006, 0.000030] | 0.000006 [0.000001, 0.000012] |
| Discretized Beta | 0.030204 [0.023471, 0.037151] | 0.002609 [0.001329, 0.004156] |

Table 1: The MSE of our estimator and the discretized-Beta estimator with both dynamic and uniform sampling strategies given synthetic problem success probabilities sampled from Uniform($[0, 1]$) for $n = 64$ problems, $k = 1\,000$, and the budget fixed at 256.

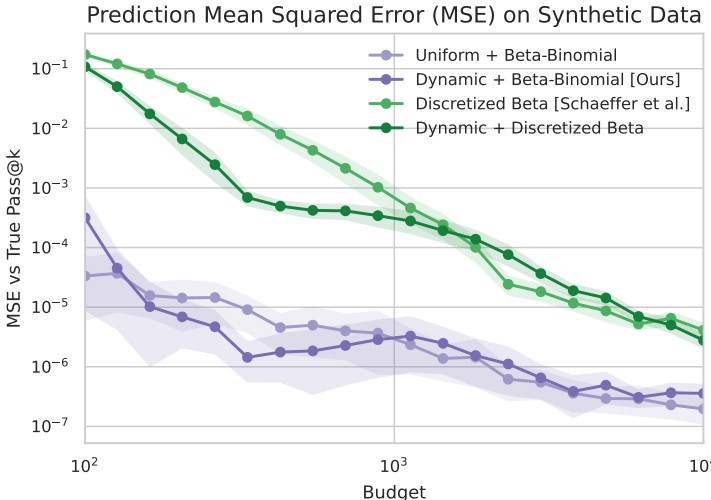

Figure 7: The MSE of our estimator and the discretized-Beta estimator with both dynamic and uniform sampling strategies given synthetic problem success probabilities sampled from Uniform$([0, 1])$ for $n = 64$ problems and $k = 1\,000$. We observe that dynamic sampling leads to modest improvements in both estimators.

## F    ADDITIONAL FIGURES

We provide matching figures from the main paper for the benchmarks that were omitted due to lack of space. Additionally, we include plots that track the scaling of mean squared error (MSE) as budget increases for fixed k, and figures containing the performance of a mixed continuous-discrete model with a discrete lump of probability at $pass@1 = 0$

## G    DATASETS

We draw our evaluation data from two recent sources: Brown et al. (2024) and Hughes et al. (2024). They record, for each of 128 prompt samples, the **number of successful outcomes out of** $10\,000$ **trials**. These prompts are sampled from three benchmark suites:

- **CodeContests** (Li et al., 2022): A competitive programming benchmark which collects description-to-code tasks from contest platforms such as AtCoder, CodeChef, Codeforces, and HackerEarth. Models are evaluated on precise correctness via test cases. Later refinements (e.g. CodeContests+) improve test case generation and validation to reduce false positives in evaluation.

- **MATH** (Hendrycks et al., 2021): A mathematical reasoning dataset of 12,500 high school competition problems (e.g. AMC, AIME). Each problem comes with a full solution path and final answer. The benchmark evaluates model proficiency in multi-step reasoning across domains such as algebra, number theory, geometry, and combinatorics.

- **AdvBench** (Zou et al., 2023): An adversarial NLP benchmark oriented toward security tasks. It emphasizes realistic attacker goals and evaluates models' success or failure under adversarial prompting strategies.

This combination lets us evaluate the efficacy of our estimator on problem success probability distributions extracted from **coding**, **mathematical reasoning**, and **adversarial robustness** domains.

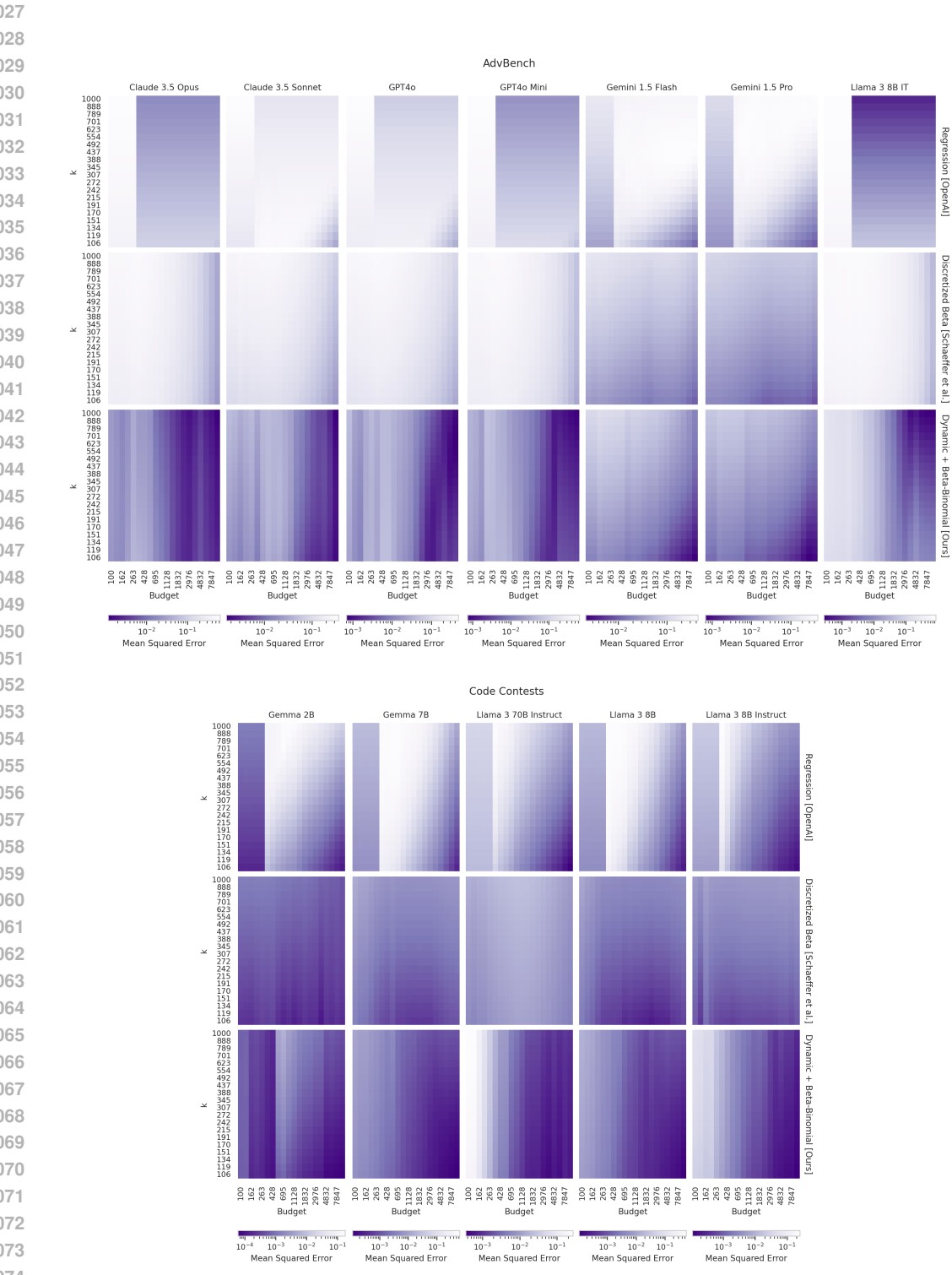

Figure 8: Heatmap depicting how predictions of $\mathrm{pass}@k$ change with the sampling budget and $k$ for MATH and Code Contests benchmarks. Note that our method outperforms existing ones for virtually all values of $k$ and sampling budget, as evidenced by the darker colors in its heatmap.

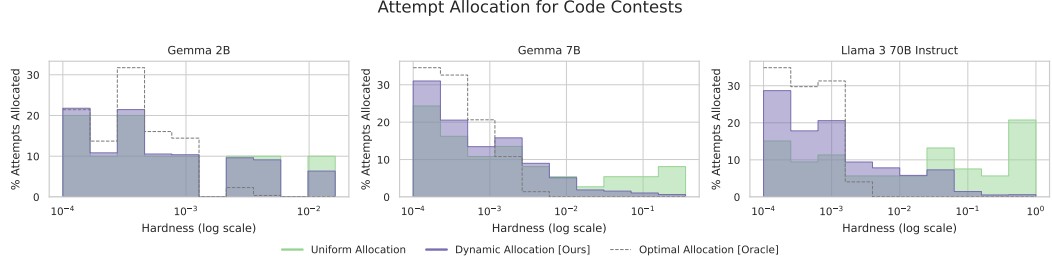

Figure 9: Contrasted distributions of problem success probabilities for the problems selected by dynamic and uniform sampling strategies on Code Contests and MATH.

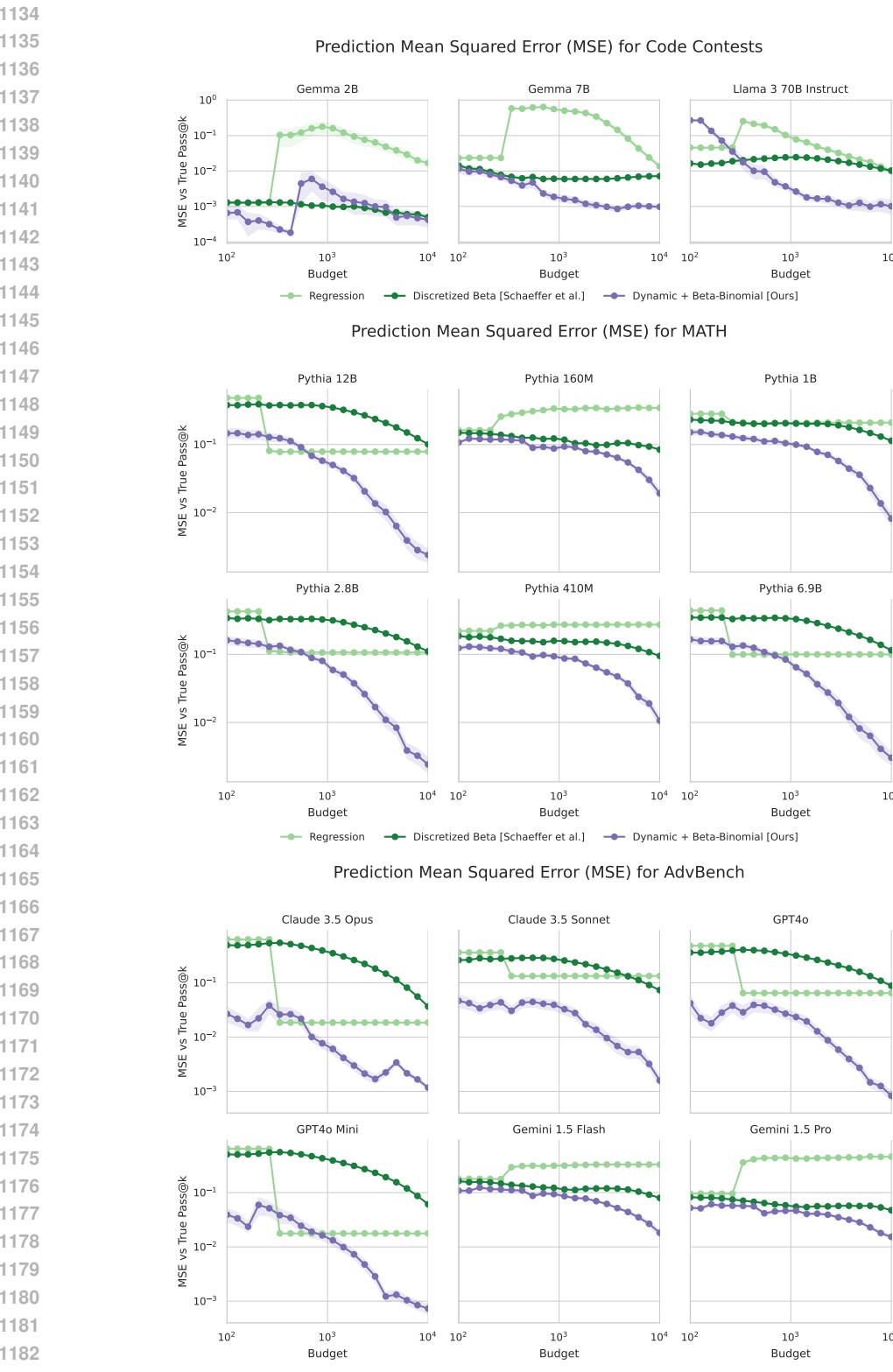

Figure 10: MSE scaling with increasing budget. As expected, more samples generally leads to a reduction in MSE across all approaches. For some models our approach reaches MSE more than 10x lower than its counterparts.

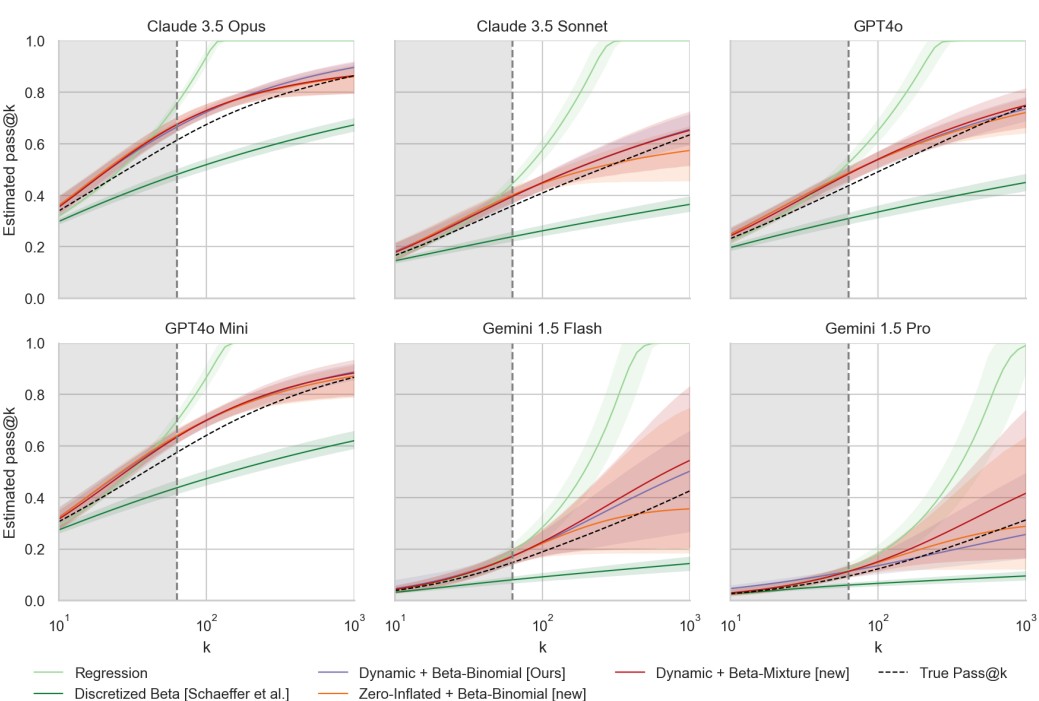

Figure 11: $pass@k$ plots comparing the performance of the mixed continuous-discrete model with a discrete lump of probability at $pass@1 = 0$ (dynamic sampling) to other methods on code contest data.

Figure 12: $pass@k$ plots comparing the performance of dynamic sampling applied to the mixed continuous-discrete model with a discrete lump of probability at $pass@1 = 0$ (Zellinger & Thomson, 2025) to other methods on BON Jailbreaking data.

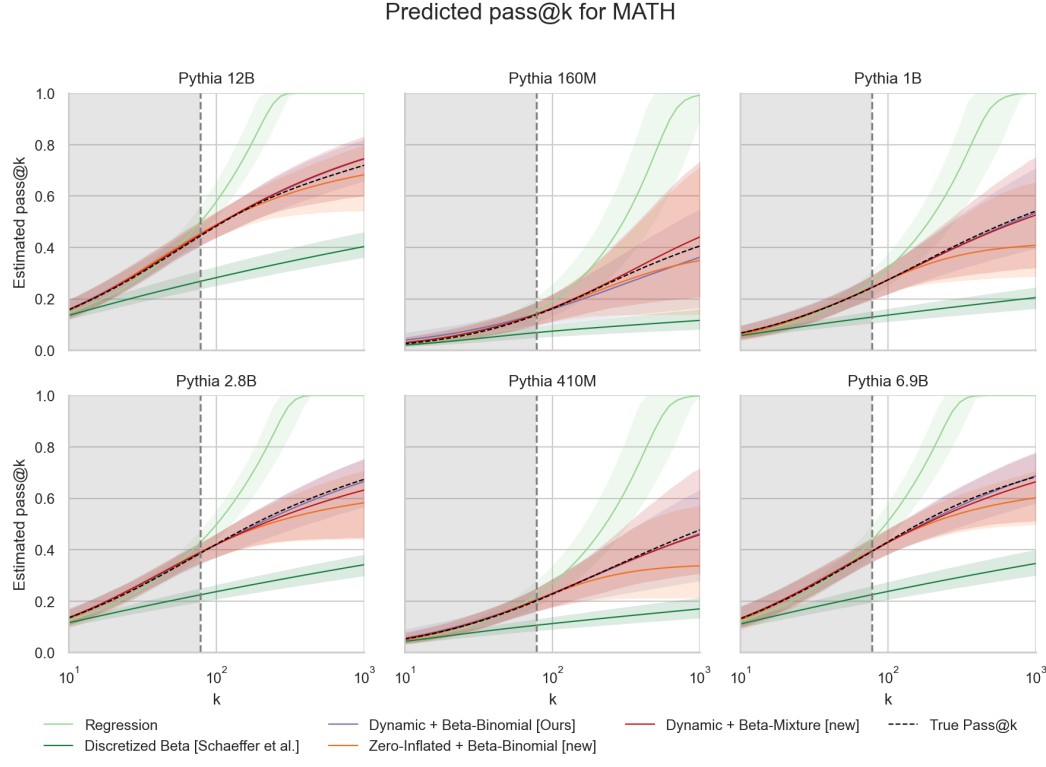

Figure 13: $pass@k$ plots comparing the performance of dynamic sampling applied to the mixed continuous-discrete model with a discrete lump of probability at $pass@1 = 0$ (Zellinger & Thomson, 2025) to other methods on MATH data.

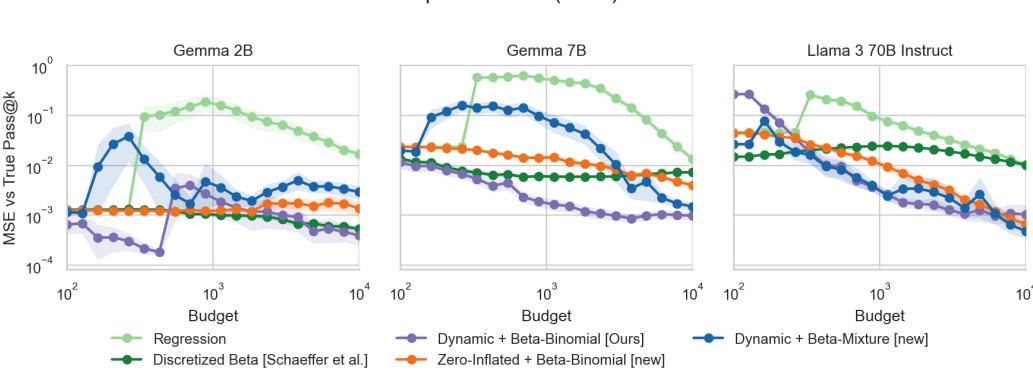

Figure 14: MSE plots comparing the performance of the mixed continuous-discrete model with a discrete lump of probability at $pass@1 = 0$ (dynamic sampling) to other methods on code contest data.

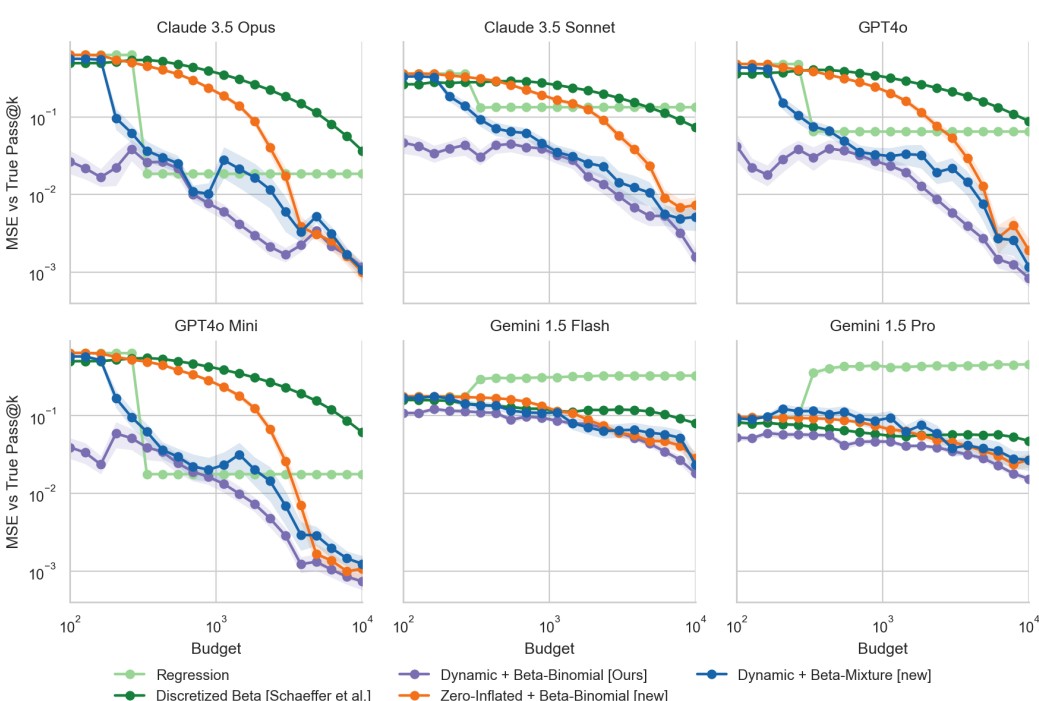

Figure 15: MSE plots comparing the performance of the mixed continuous-discrete model with a discrete lump of probability at $pass@1 = 0$ (dynamic sampling) to other methods on jail-breaking data.

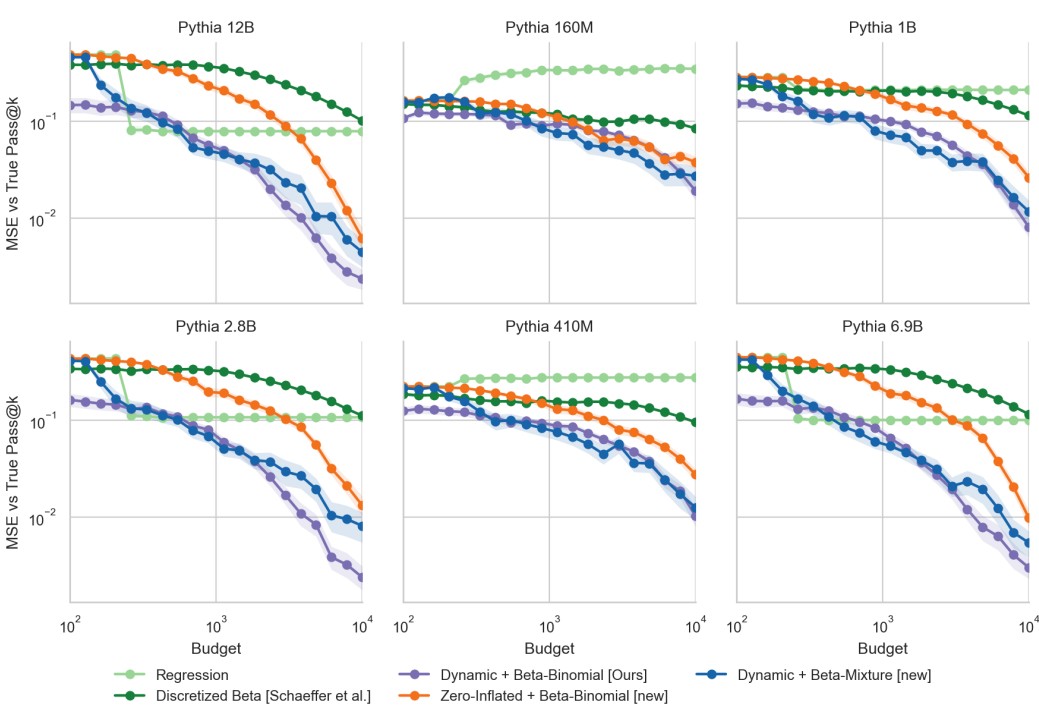

Figure 16: MSE plots comparing the performance of the mixed continuous-discrete model with a discrete lump of probability at $pass@1 = 0$ (dynamic sampling) to other methods on math data.

