# OpenReview forum: "Efficient Prediction of pass@$k$ Scaling\\in Large Language Models"
_ICLR.cc/2026/Conference — Submitted to ICLR 2026_

### Official Review · Reviewer_CBht · 2025-10-18

**Soundness:** 3
**Presentation:** 4
**Contribution:** 3
**Rating:** 4
**Confidence:** 3

**Summary:**

This paper aims to efficiently and accurately predict model performance across massive repeated queries using only a limited number of trials. The authors 1) identify the limitations of two existing approaches for pass@$k$ estimation, i.e., linear regression and distributional fitting; 2) propose an alternative fitting method and a dynamic sampling strategy; and 3) demonstrate that the proposal brings significant improvements across various hyperparameters and datasets.

**Strengths:**

1. The motivation and the research problem are clearly introduced at the beginning of the paper. The problem is also significant for the practical deployment of LLMs at scale.
2. The authors contribute to the problem in multiple ways, including a formal theoretical review and the proposal of targeted solutions.
3. The experimental results are well presented and demonstrate that the proposed method outperforms the baselines by a clear margin.

**Weaknesses:**

1. Given that $k<B$, i.e., dozens to thousands of attempts, I don't think the setup corresponds to what is mentioned in L40, i.e., billions of daily interactions.
2. The authors list the $k$-dependency as a drawback of linear regression for pass@$k$ estimation. However, as pass@$k$ inherently depends on $k$, there should be a more detailed justification for this claim.
3. In this paper, both linear regression and distributional fitting are criticized for being inapplicable to smaller $k$. While I agree that a robust scaling law should generalize across different values of $k$, the authors’ claim that these pass@$k$ predictions are most relevant to large-scale deployment may suggest a mismatch between the critique and their stated motivation.
4. The discussion section feels somewhat brief. Since you have pointed out the limitations of existing estimation approaches and designed your algorithm accordingly, I recommend you to provide a more detailed analysis clarifying how your proposals contribute to the efficiency and accuracy, and whether your method avoids the aforementioned limitations.

**Questions:**

1. For the mention of success rate in L36, I think the typical interpretation of this metric is that it's averaged across all attempts. You may refer to the definitions of empirical probability and expected maximum toxicity in the RealToxicityPrompts paper (Gehman et al., 2020) for a more rigorous statement.
2. Could you provide examples of the larger $k$ values mentioned in L194?
3. The proposed method seems to be a more flexible and efficient version of the distributional fitting approach. If my understanding is correct, I would like to see the individual contributions of these two improvements.
4. Could you elaborate on why you chose these ranges for $k$ and $B$ and how the conclusions drawn from this setting would inform the model providers and regulators?

---

> ### Author Response · Authors · 2025-11-20
> **Rebuttal to Reviewer CBht**
>
> Thank you for the careful and thorough review.  We try to address your concerns below.
>
> **Weaknesses:**
>
> > [...] I don't think the setup corresponds to what is mentioned in L40, i.e., billions of daily interactions.
>
> While our absolute k values are necessarily smaller than those in billion-interaction deployments, the underlying statistical challenge is identical: in both settings, one must extrapolate several orders of magnitude beyond the available sampling budget. Predicting pass@10,000 from ~100 samples is the same structural problem as predicting pass@1M from ~10k samples. Our methodology targets this extrapolation challenge rather than any specific absolute scale.
>
> > The authors list the k-dependency as a drawback of linear regression for pass@k estimation. However, as pass@k inherently depends on k, there should be a more detailed justification for this claim.
>
> We respectfully clarify this as a misunderstanding.  Dependence on $k$ is not a problem when computing different estimates of pass@k: indeed, estimates of pass@$k$ must depend on k.  There are two other quantities that problematically depend on k.  First, the _variance_ in estimation of pass@$k$ depends on k in Chen et. al.’s prediction method, meaning that the data inherently violates the regression assumption that estimation noise is independent of the covariates.  Second, estimates for pass@$k$ and pass@$k’$ are not independent for $k\neq k'$, which violates the regression assumption that the data are i.i.d.  We prove both of these at some length in Appendix A.  While counterfactuals do not exist that allow us to isolate the exact effect of each of these violations of the regression assumptions, they should make one wary of predictions that come out of the regression model.  Indeed, the predictions from the regression model are poor compared to all distributional fitting methods.
>
> >mismatch between the critique and their stated motivation.
>
> To collect 10,000 samples per problem, it required 1.28 million samples per dataset and model.  Scaling up even one order of magnitude would be infeasible for us.  The main challenge that industry labs face is predictions across orders of magnitude, so while we cannot match the exact orders of magnitude, we can simulate the challenges faced by crossing several orders.  For instance, an industry lab may need to predict what will happen with 1M samples per problem using 10,000 samples per problem, and we try to predict what happens with 10,000 samples per problem using 100 samples per problem.  As we stated before, while the tasks are not identical, the lessons learned could be transferable to higher orders of magnitude.
>
> >The discussion section feels somewhat brief. Since you have pointed out the limitations of existing estimation approaches and designed your algorithm accordingly, I recommend you to provide a more detailed analysis clarifying how your proposals contribute to the efficiency and accuracy, and whether your method avoids the aforementioned limitations.
>
> Thanks for the suggestion.  We have revised the conclusion in the updated version of the paper to explain how we use valid statistical methods like MLE on an exponential family model rather than fitting models on top of correlated estimators.
>
> **Questions**
>
> 1.  We had a look at Gehman et. al. 2020.  We are referring to the fact that pass@k (for the full dataset) quickly approaches 100%, even for models that had very low pass@1 success rates, as in Brown et. al. 2024.
> 2.  The problem already shows up when trying to predict pass@1000, as shown in Figure 1.  The heatmap in Figure 5 shows MSE when predicting pass@10000, and for past methods one can see that the MSE is much higher than for ours.
> 3.  Please see Figure 7, which we have added to the Appendix.  Most of our gains come from switching to MLE for optimizing an unscaled beta-binomial.  The dynamic sampling method only helps with certain distribution types (though it never tends to hurt performance).  Figure 4 explores which distributions the dynamic sampling method is effective for.  Figure 6 shows an example where the dynamic sampling method has a large impact on performance (holding the  fitting method constant).
> 4.  We chose $B$ between 1/100th and 1/10th of our total data, since we wanted to cross more than an order of magnitude during prediction.  We chose the largest ranges for $k$ that our datasets allowed.  Industry labs face a similar problem on a larger scale. They have a fixed compute budget and must estimate quantities similar to pass@$k$ where $k$ is several orders of magnitude larger than what they can directly estimate.  We cannot match the scale of industry labs, but we believe that some of the lessons about dynamic sampling and statistical fastidiousness transfer.
>
> Thanks again for your review; we appreciated your suggestions.  If we have addressed your concerns, please consider raising your score.

---

> > ### Comment · Reviewer_CBht · 2025-11-27
> >
> > Thanks for the clarifications and the additional experimental results. I would like to maintain my score at this point.

---

> ### Author Response · Authors · 2025-11-27
>
> Thank you for engaging with our rebuttal and updated experiments.
>
> To summarize, we clarified:
>
> (i) that our goal is to study extrapolation across orders of magnitudes rather than to make predictions at any particular absolute deployment scale
>
> (ii) why regression-on-pass@$k$ violates key assumptions
>
> (iii) the contribution of the MLE fit from the dynamic sampling strategy with new experiments.
>
> If there are still specific issues that prevent a higher score, we would be grateful if you could indicate what these are for the AC's context and to help us improve the paper going forward.

---

> > ### Comment · Reviewer_CBht · 2025-11-28
> >
> > I appreciate the ablation analysis in Fig. 7, and now I understand that the k-dependence itself is not a drawback of regression-on-pass@k. The reason for maintaining my score is that I remain unsure whether the lessons learned can be broadly applied. In other words, while I agree that this paper addresses the cross-order problem very effectively, I am not fully convinced that the conclusions generalize, leaving some potential for over-claiming. Since the scoring options are only 4 and 6, I would like to choose 5.

---

### Official Review · Reviewer_5fkD · 2025-10-25

**Soundness:** 4
**Presentation:** 3
**Contribution:** 4
**Rating:** 6
**Confidence:** 3

**Summary:**

The paper tackle how to accurately predict pass@k for large k when you only have a small sampling budget. They show why popular approaches (e.g., log–log power-law regression and a discretized beta fit) give biased, high-variance extrapolations, and then propose to replace them with a beta–binomial fit to per-problem success rates and a dynamic policy that concentrates trials on the hardest problems, then shows this combo matches ground truth far better across AdvBench, MATH, and Code Contests benchmarks.

**Strengths:**

The paper:
- Investigates a valuable, high-impact problem: how to predict pass@k under tight sampling budgets, which matters for safety (scaling of rare failures) and capabilities (planning compute for methods like RLVR).

- Provides a clear critique of existing approaches: shows why log–log regression and discretized-beta fits are statistically flawed and yield poor high-k extrapolations.

- Introduces a novel solution with strong empirical evidence: a beta–binomial estimator plus dynamic sampling that focuses trials on harder items; across AdvBench, MATH, and Code Contests, it tracks ground truth much better than baselines.

- Stress tests & ablations clarify when it helps most: allocate more samples to hard problems, and in heavy-tailed cases, dynamic sampling usually beats (or at least matches) uniform across distributions.

**Weaknesses:**

- Certain assumptions in the proposed solution, that is i.i.d. attempts with a fixed per-problem success rate, a single Beta prior over difficulty, and assuming task stationarity (no changes in guardrails, prompts, or caching) over time, may reduce robustness of the proposed method and introduce bias.
- Evaluations cover a few benchmarks and ~hundreds of items. Although useful but limited for broad generalization.
- The paper primarily validates the methods using MSE to ground-truth pass@k curves, which is limited. Incorporating calibration measures and decision-centric metrics (e.g., risk at a target pass@k) would give a better picture of the proposed method’s advantages.
- Limited number of baselines considered for comparison. Adding richer-prior baselines, e.g., Beta mixtures or Dirichlet-process priors that better handle tail misspecification, would more clearly demonstrate the proposed method’s advantages.

**Questions:**

- Could you clarify why the error bars differ so much across LLMs, especially for the Gemini models compared with the others?
- What happens under non-stationarity (model updates, prompt/guardrail changes)? Do you have a rolling or re-weighting variant that stays calibrated?
- How does your method handle non-i.i.d. retries on the same problem. For example, when later attempts reuse earlier chain-of-thought or tool outputs (adaptive changes in success probability) or when samples are correlated (e.g., beam/diverse decoding sharing prefixes)?

---

> ### Author Response · Authors · 2025-11-20
>
> Thank you for the diligent review and for highlighting the importance of accurately predicting pass@k under tight sampling budgets.  We address the weaknesses that you flagged below.
>
> **Weaknesses**
>
> >Certain assumptions in the proposed solution [...] may reduce robustness of the proposed method and introduce bias.
>
> This is an interesting point, and it is currently the subject of a follow-up paper that we are writing.  For pre-release model testing and scaling inference-time compute, stationarity is a valid assumption.  When generating rollouts for RLVR, we agree that there could be some evolution over time, but resampling thousands of rollouts after every training step to test for this evolution is computationally expensive, so using the initial distribution as a proxy to estimate the cost of solving particular problems is a computationally feasible alternative.
>
> >Evaluations cover a few benchmarks and ~hundreds of items. Although useful but limited for broad generalization.
>
> Evaluations actually involve 10000 samples per problem to serve as a proxy for ground truth.  We try to estimate pass@10000 using dozens to hundreds of samples.  Testing 128 prompts with 10000 samples each already requires 1.28 million samples per model and dataset, so scaling up enough to extend results meaningfully on a log(pass@k) scale would be prohibitively expensive for us.  We believe that the lesson of how to cross several orders of magnitude in prediction could still be transferrable to larger scales.
>
> >The paper primarily validates the methods using MSE to ground-truth pass@k curves, which is limited. Incorporating calibration measures and decision-centric metrics (e.g., risk at a target pass@k) would give a better picture of the proposed method’s advantages
>
> We interpret ‘risk at a target pass@k’ as the probability that the model underperforms a required pass@k threshold (i.e., the tail risk of the predictive distribution). Under this interpretation, our beta–binomial framework already produces a posterior distribution over pass@k, which can be used to compute such risk directly (unlike regression-based methods).
>
> > Limited number of baselines considered for comparison. Adding richer-prior baselines, e.g., Beta mixtures or Dirichlet-process priors that better handle tail misspecification, would more clearly demonstrate the proposed method’s advantages.
>
> We tested Beta mixture priors optimized using EM when writing the paper and found that they offered no advantage over a single beta prior.  We have added graphs with these tests to Figures 11, 12, and 13 in the appendix.
>
> **Questions**
>
> > Could you clarify why the error bars differ so much across LLMs, especially for the Gemini models compared with the others?
> We compute error bars for each of our estimators by bootstrapping the 2.5th and 97.5th percentile MSE across 100 runs over 1000 resamplings.
>
> Generally, we should expect that some distributions of problem difficulty are more difficult to estimate despite yielding the same overall pass@k. Consider for instance, the following two LMs which both have a true pass@100 of 0.7:
>
> LM A:
> - 30% of tasks: never solves
> - 70% of tasks: always solves
>
> LM B:
> - 100% of tasks: solves with probability ~0.356%
> Treating individual tasks as inscrutable black boxes, aleatoric uncertainty is significantly higher for LM B, resulting in higher variance estimates. We see a similar pattern when comparing the Gemini 1.5 Flash to Claude 3.5 Opus on AdvBench: Claude is jailbroken on over 60% of tasks with probability greater than 1% while for Gemini this number is closer to 10%.
>
> > What happens under non-stationarity (model updates, prompt/guardrail changes)?
>
> This is a very interesting question!  The data sources that we have available do not contain updates over time or dynamic sampling methods, and it would be highly expensive to collect enough new samples after each step to measure such a distribution shift.  Although this question is outside the scope of this particular paper, follow-up work could learn, for instance, a parameterization of $\alpha(t), \beta(t)$ in the beta binomial rather than single values.  In order for the dynamic sampling method to remain useful under distribution shift, some theoretical conditions would likely be required, such as a monotonic increase of success probabilities over time.
>
> In the context of beam/diverse decoding, the beta-binomial model is misspecified.  One could potentially replace the binomial component of our current model with a model from survival analysis (assuming a greater probability of success at each step).  Modelling sequential sampling methods is the subject of a follow-up paper that we are working on currently.
>
>
> We appreciate the insightful comments and suggestions. If we have sufficiently addressed your concerns, we hope you will consider raising your score.

---

> > ### Comment · Reviewer_5fkD · 2025-11-25
> > **Reviewer Response**
> >
> > Thank you for your response, I truly appreciate the effort you put into addressing my questions and concerns. I will keep my scores.

---

> ### Author Response · Authors · 2025-11-27
>
> Thank you again for engaging with our rebuttal.
>
> Just to briefly summarize how we updated the paper in response to your comments:
>
> - We clarified the regimes where the approximate stationarity assumption is appropriate (e.g., pre-release evaluation and scaling inference-time compute) and explicitly discussed its limitations in RL/rollout settings, as well as potential extensions to time-varying scenarios.
> - We emphasized that our beta–binomial framework yields a full posterior over pass@$k$, which directly supports decision-centric quantities such as "risk at a target pass@$k$" (tail risk of underperforming a required threshold), complementing the MSE-based comparisons.
> - We added experiments with richer-prior baselines (beta mixtures fit via EM) to Figures 11–13, and found that they do not improve over a single beta prior in our benchmarks.
> - We elaborated on the discussion of error bars and variance across models, including intuition for why certain difficulty distributions intrinsically lead to higher estimator variability, even at similar overall pass@$k$.
>
> We fully understand that you are keeping your scores, but if there are still specific concerns that you feel prevent a higher overall assessment, we would be grateful for a brief indication.

---

### Official Review · Reviewer_JNYc · 2025-10-31

**Soundness:** 3
**Presentation:** 3
**Contribution:** 2
**Rating:** 4
**Confidence:** 2

**Summary:**

Current LLMs have a large number of users and their behavior in such conditions is not well studied. In particular, trying to understand performance for “pass @ k” where k is a large number is important. As such, this paper tries to predict the performance of models for “pass @ k” where k is large and experiments cannot be run to provide exact measurements. The authors model task success probabilities with a beta-binomial distribution, which allows accurate extrapolation from limited data. They also propose a dynamic sampling strategy that focuses more on hard tasks, improving prediction accuracy and efficiency. Overall, their method provides a practical, data-efficient way to estimate LLM performance or failure rates at scale.

**Strengths:**

A simple method for estimating the performance of models for pass @ k

**Weaknesses:**

Not clear how the method performs for tasks that are subjective.

**Questions:**

Do you have an intuition how well the method performs for tasks that involve safety, which tend to be subjective in nature (as opposed to the math/reasoning tasks you’re covering in the paper)?

---

> ### Author Response · Authors · 2025-11-16
> **Rebuttal to reviewer JNYc**
>
> Thanks for taking the time to read and review our paper.  We address the weakness and concern that you flagged below.
>
> >Not clear how the method performs for tasks that are subjective.
>
> Pass@k is a metric used for tasks with verifiable rewards.  The rewards can be verified using a language model, which allows pass@k to be adapted to “subjective tasks”.  The need for verifiable rewards could be cast as a limitation of pass@k as a metric, but probably not as a limitation of our paper, which estimates pass@k.
>
> > Do you have an intuition how well the method performs for tasks that involve safety, which tend to be subjective in nature (as opposed to the math/reasoning tasks you’re covering in the paper)?
>
> Roughly a third of the experiments in this paper involve safety applications, for which we used the jailbreaking dataset from [1].  Harmful responses were first identified by a classifier and then verified by a human.
>
> If we have addressed all of your concerns, we hope that you’ll consider raising your score.
>
> [1] https://arxiv.org/abs/2412.03556

---

> > ### Comment · Reviewer_JNYc · 2025-11-21
> >
> > Thank you for the clarification. I will maintain my scores.

---

> > > ### Author Response · Authors · 2025-11-23
> > > **Response to Reviewer JNYc**
> > >
> > > Thank you for your response. Since our rebuttal addressed your primary stated concern regarding safety tasks (by highlighting our experiments on AdvBench/jailbreaking shown in Figure 1 and Figure 3), it is unclear what specific deficiencies remain to justify a rejection.
> > >
> > > Could you please clarify which outstanding technical weaknesses are preventing you from raising the score? Providing this justification is crucial for us to understand your assessment.

---

### Official Review · Reviewer_uMij · 2025-11-01

**Soundness:** 3
**Presentation:** 3
**Contribution:** 3
**Rating:** 4
**Confidence:** 4

**Summary:**

The authors tackle the problem of predicting LLMs' pass@k for verifiable problems, for a specific regime where the budget of LLM calls (for gathering data to predict pass@k) is fixed and k exceeds the average LLM budget per problem in the data set, thereby necessitating extrapolation. The authors propose making variable numbers of attempts per query by spending more of the LLM budget on problems with the low success rates. They present a theorem stating that if the probability of success is known exactly for each problem, difficult problems should receive a greater share of the budget for LLM calls, validating their approach.

**Strengths:**

- The paper is well written and interesting.
- The dynamic sampling strategy is sound and makes perfect sense for the problem: this is a valuable contribution to the literature.
- The theorem justifying over-sampling of difficult problems is mathematically clean.

**Weaknesses:**

- The framing of the significance of the problem seems misleading. For example, the cited research on brute-force jailbreaking actually varies the prompt on each attempt, so it's not directly comparable to repeated sampling. Similarly, it is not clear that "The relevance of [predicting pass@k] is only underscored by the massive scale at which these frontier AI systems are deployed, with some experiencing billions of daily
interactions." - does massive scale imply that customers are running the same query a million times? How does the rise of reasoning LLMs affect the significance of pass@k: given the increased latency of producing a thinking trace, does it still make sense to sample repeatedly from these models?
- Pass@1 is a fraught quantity: either an LLM is capable of answering a question, in which case it will get the correct answer reasonably quickly after some repeated trials, or the problem is beyond the LLM's capability. In the latter case, the probability of answering correctly is near zero. However, correctly estimating pass@1 for difficult problems with very low but nonzero success rates is important for accurately predicting pass@k as k -> inf. It appears that estimating pass@1 values with greater precision is the main benefit of your dynamic sampling approach, NOT the reduced variance of the pass@k estimator as described in your theorem (although that is an added benefit). The theorem assumes that pass@1 is known precisely, whereas in reality, estimating pass@1 seems to be the core problem, diminishing the significance of your theorem.
- Your evaluation estimates each problem's ground truth pass@1 with only 10,000 samples, effectively capping the resolution of detecting nonzero pass@1 at somewhere near 1e-4. Is this justified? I could easily see true pass@1 value to be smaller than 1e-4. This resolution limit affects all your posted results, so a discussion would be valuable. Do you expect many pass@1 values to be smaller than 1e-4 in practice?
- Your beta model for estimating the distribution of pass@1 may not adequately capture the heavy concentration of probability mass near zero. Perhaps a mixed continuous-discrete model with a discrete lump of probability at pass@1 = 0 would be a sensible approach. For example, see the paper https://openreview.net/forum?id=YCBVcGSZeR.
- This paper is not "scaling law research," which seems to imply a connection to the famous scaling laws from the pre-training literature. This paper is about pass@k and repeated sampling from LLMs. Thus, the related work section should not cover "scaling laws" in the abstract but stick to the paper's core question of repeated sampling from LLMs. Specifically, it would be relevant to discuss the other side of the coin of repeated-sampling: non-verifiable problems, where majority voting etc. must be applied. For example, see the paper https://openreview.net/forum?id=m5106RRLgx. Similarly, statistics papers on estimating very small probabilities could be relevant. As it stands, the positioning of the paper within the literature appears misleading and the related work section should stick closer to the knitting of the paper: repeated LLM calls.

**Questions:**

- Please see "Weaknesses". Overall, I respect your paper a great deal and consider your dynamic sampling methodology to be an important contribution to the literature. That said, I would appreciate more discussion or clarification on the issue of very small pass@1 values, and how to interpret your evaluation in light of such issues. I'm still unsure if the paper meets ICLR's bar in terms of substantial novelty and significance, and welcome your comments.
- Upon introducing Algorithm 1, I'd recommend clarifying that the arrays of "attempts" and "successes" are mutable arrays that start out empty and will be gradually extended (during Algorithm 2). As written, Algorithm 1 seems to require the input arrays to be fully formed. It also seems that you could describe Algorithm 1 more simply: essentially, the idea is to sample among all problems that have not yet observed a success and within all such "difficult" problems, the least-attempted ones are prioritized. Correct? I have this impression because in practice, the pool of problems with zero successes will likely never shrink to zero, since there will be SOME pass@1 values low enough never to yield a success.
- Some empirical illustration of the scale and magnitude of typically observed pass@1 values (specifically the ones near zero) would be appreciated.

---

> ### Author Response · Authors · 2025-11-18
> **Rebuttal to uMIJ**
>
> We thank the reviewer for their thorough evaluation and insightful questions. We are particularly encouraged by the recognition that our dynamic sampling strategy is 'sound and makes perfect sense' and represents 'a valuable contribution to the literature.' We address each concern below.
>
> **Weaknesses**
> > does massive scale imply that customers are running the same query a million times?
>
> In some cases, like startups calling LM APIs, yes.  In our jailbreaking data, the harmful prompts are slightly varied, as in [1], and we can still apply the pass@k metric to the marginal distribution (assuming a randomly chosen jailbreak against a given problem has a marginal probability of success p).  In GRPO, the same prompts are also sampled dozens to thousands of times.
>
> > The theorem assumes that pass@1 is known precisely, whereas in reality, estimating pass@1 seems to be the core problem, diminishing the significance of your theorem.
>
> We think that there are two misconceptions here.  First, one can decompose the least squares risk in estimation problems into a bias and a variance term.  Reducing the variance of an estimator will improve its risk.  The point of the theorem is not that pass@1 values are known, but that the optimal budget that one should allocate to a given question is asymptotically a function of its difficulty.  Although we don’t know the exact pass@1 values as we are sampling, the theorem says that we should allocate more of the budget to problems that are likely to have lower pass@1 values, which during sampling are the problems for which we have fewer correct answers.  Figure 3 shows that our dynamic sampling allocates the sampling budget more in line with the optimal distribution.
>
> >Do you expect many pass@1 values to be smaller than 1e-4 in practice?
>
> We have added a limitations section to address this point.  Certainly we expect some values of pass@k to be orders of magnitude smaller.  Unfortunately, collecting huge numbers of samples costs thousands of dollars, so scaling up the number of samples by even 1 more order of magnitude would not have been feasible for our budget. This is a problem that companies also face when assessing deployment risk– they may have budgets of 100k samples per harmful prompt, but users will potentially test a model’s vulnerabilities/capabilities tens of millions of times.  Therefore, our setting still reflects (albeit on a smaller scale) the realities that frontier labs face when crossing orders of magnitude to make predictions.
>
> >Your beta model for estimating the distribution of pass@1 may not adequately capture the heavy concentration of probability mass near zero.
>
> We implemented this method: results can be found in Figures 14-16 of the updated manuscript in the appendix.  When we place a fixed probability mass at zero (optimized again by using the EM algorithm), the models are too pessimistic in the MATH settings, and they lose their power law scaling at high $k$.  The fits are similar to our current ones but look slightly worse at high values of $k$.
>
> > This paper is not "scaling law research.”
>
> This is a fair point.  We have rewritten the related work section to focus specifically on pass@k, citing the paper that you mentioned.
>
> **Questions**
> >I'm still unsure if the paper meets ICLR's bar in terms of substantial novelty and significance, and welcome your comments.
>
> We think that rigorously studying past methods and explaining where and why they fail is a valuable contribution to the field.  We cut squared prediction error losses, sometimes by more than an order of magnitude, across a variety of models and settings.  We support our insights with theorems and careful mathematical study that could save empiricists time and compute.  Given the current lack of rigor and careful analysis in ML research, we think that it is important for papers like this to appear in ICLR.
>
> >Upon introducing Algorithm 1, I'd recommend clarifying that the arrays of "attempts" and "successes" are mutable arrays that start out empty and will be gradually extended (during Algorithm 2).
>
> You make some good points.  We specifically wrote the algorithm with immutable arrays for fixed datasets of questions (which is what many researchers use today), but we agree that there’s an easy extension to infinite possible question sets as long as the questions arrive in a stream (so there’s a finite pool of questions at any given time).  We have noted this in Section 4.2 of the manuscript.  Your point about the number of problems with 0 successes never shrinking to 0 was true in the datasets we examined, but it generally depends on the model-dataset pairing.
>
>
> > Some empirical illustration of the scale and magnitude of typically observed pass@1 values (specifically the ones near zero) would be appreciated.
>
> We use the same data as [2]. We did not include these graphs because they can be found in Figure 4 of [2].
>
>
> [1]  https://arxiv.org/abs/2412.03556
>
> [2] https://arxiv.org/pdf/2502.17578

---

### Author Response · Authors · 2025-12-01

Dear AC,

We would like to take a moment to highlight the most substantial reviewer concerns about our paper and how we addressed them in the rebuttal.

1.  **Framing and scale (uMij, CBht).** uMij and CBht asked us to better clarify how our setting relates to large-scale deployment and to “scaling laws.” We rewrote the related work, and edited the limitations and the introduction to frame the paper explicitly as cross-order extrapolation (e.g., predicting pass@10,000 from around 100 samples).  We removed pre-training “scaling law” language, and focused the positioning squarely on repeated sampling and pass@k. We now explicitly note that, while our absolute k is smaller than what frontier labs see, the statistical problem is identical: an industry lab might need to predict pass@1M from around 10k samples per prompt, while we predict pass@10k from about 100. In both cases, one must extrapolate several orders of magnitude beyond the available sampling budget.

2.  **Extremely small pass@1 and mass at zero (uMij).** uMij questioned whether 10k samples per problem and a continuous Beta prior can capture very small success probabilities, and suggested adding a point mass at 0. We added a limitations section acknowledging the finite resolution and cost constraints, implemented the suggested mixed continuous–discrete model with a mass at zero, and found it to be overly pessimistic and to fit our benchmarks worse than the simple beta–binomial; these results are now included in the appendix.

3.  **Assumptions, richer priors, and decision-centric metrics (5fkD).** 5fkD highlighted stationarity/i.i.d. assumptions, the focus on MSE, and asked for richer priors. We clarified when stationarity and i.i.d. are reasonable and where they are limitations, flagged non-stationary/adaptive settings as follow-up work, added Beta-mixture baselines fit via EM (which did not improve over a single Beta on our benchmarks), and emphasized that the beta–binomial yields a full posterior over pass@k, supporting decision-centric quantities such as the risk of underperforming a target pass@k.

4.  **Safety / “subjective” tasks (JNYc).** JNYc asked how the method applies beyond math/reasoning tasks. We clarified that roughly one third of our experiments are safety-oriented (AdvBench/jailbreaking) with classifier + human verification, and that the requirement of verifiable rewards is a property of the pass@k metric itself rather than a limitation of our estimator.


Reviewers generally agree that we address an important deployment-relevant problem, provide a clean analysis of why existing extrapolation methods fail, and introduce a simple beta–binomial + dynamic sampling approach that clearly improves pass@k prediction across AdvBench, MATH, and Code Contests. One reviewer scores the paper 6 (above threshold); the others score 4, with one stating they would choose 5 if that option existed. In light of this and the additional experiments and clarifications, we believe the paper meets the ICLR bar and respectfully advocate for acceptance.  Our paper is a careful mathematical analysis that identifies flaws with past methods, corrects them, and thereby achieves better results.  This type of work has become increasingly uncommon in recent years; we believe that this exhibition of performance gains through mathematical rigor is an important contribution to the field, especially given the recent shift away from mathematical principles and towards purely empirical research.

---

### Meta-Review · Area_Chair_7RLx · 2025-12-23

**Summary:**

The paper studies the problem of predicting LLMs' pass@k for a small budget of LLM calls and large k. Overall, the reviewers appreciated the studied problem and liked the principled statistical approach taken in this paper.

However, the paper received limited support for acceptance. Concerns include the usage (and estimation of) pass@1 (Reviewer uMij), strong assumptions such as i.i.d. attempts with a fixed per-problem success rate (Reviewer 5fkD) and beta prior (Reviewers 5fkD and uMij) and applicability to the large-scale settings used for motivation (Reviewer CBht).

Overall, the paper seems promising, but the concerns indicate that the work does not currently meet the bar for acceptance. In particular, the authors are encouraged to further discuss (and possibly relax) their modelling assumptions and to consider whether and how their method can be shown to extend to large-scale settings.

**Reviewer Concerns:**

The authors responded by stressing the merits of solving the extrapolation problem for smaller regimes as a counterpart for the motivating LLM problem. They also argued why some of their modelling assumptions are suitable and added a discussion on limitations.

Overall, the response clarified the assumptions made. However, it remains unclear if this set of assumptions is suitable for the presented motivation of this work.

**Reviewer Scores:**

3/4 reviewers responded and said they will keep their score (more precisely, Reviewer CBht stated they would raise to 5). Reviewer uMij did not respond before the end of discussions. However, some of their concerns are shared with other reviewers (e.g. about the suitability of considering this method in the context of large-scale LLM usage and about the beta prior), who did not raise. Therefore, I believe it is unlikely that reviewer uMij would have raised substantially (e.g. to more than 6).

---

### Decision · Program_Chairs · 2026-01-26

Reject